# Assessing the danger of self-sustained HIV epidemics in heterosexuals by population based phylogenetic cluster analysis

Teja Turk[1,2], Nadine Bachmann[1,2], Claus Kadelka[1,2], Jürg Böni[2], Sabine Yerly[3], Vincent Aubert[4], Thomas Klimkait[5], Manuel Battegay[6], Enos Bernasconi[7], Alexandra Calmy[8], Matthias Cavassini[9], Hansjakob Furrer[10], Matthias Hoffmann[11], Huldrych F Günthard[1,2†], Roger D Kouyos[1,2†‡*], Swiss HIV Cohort Study

[1]Division of Infectious Diseases and Hospital Epidemiology, University Hospital Zurich, Zurich, Switzerland; [2]Institute of Medical Virology, University of Zurich, Zurich, Switzerland; [3]Laboratory of Virology, Geneva University Hospitals, Geneva, Switzerland; [4]Division of Immunology and Allergy, University Hospital Lausanne, Lausanne, Switzerland; [5]Molecular Virology, Department of Biomedicine - Petersplatz, University of Basel, Basel, Switzerland; [6]Division of Infectious Diseases and Hospital Epidemiology, University Hospital Basel, Basel, Switzerland; [7]Division of Infectious Diseases, Regional Hospital Lugano, Lugano, Switzerland; [8]Division of Infectious Diseases, Geneva University Hospitals, Geneva, Switzerland; [9]Service of Infectious Diseases, Department of Medicine, Lausanne University Hospital, Lausanne, Switzerland; [10]Department of Infectious Diseases, Bern University Hospital, University of Bern, Bern, Switzerland; [11]Division of Infectious Diseases, Cantonal Hospital St. Gallen, St. Gallen, Switzerland

*For correspondence: roger. kouyos@usz.ch

†These authors contributed equally to this work

Present address: ‡Division of Infectious Diseases and Hospital Epidemiology, University Hospital Zurich, Zurich, Switzerland

Group author details: Swiss HIV Cohort Study See page 14

**Abstract** Assessing the danger of transition of HIV transmission from a concentrated to a generalized epidemic is of major importance for public health. In this study, we develop a phylogeny-based statistical approach to address this question. As a case study, we use this to investigate the trends and determinants of HIV transmission among Swiss heterosexuals. We extract the corresponding transmission clusters from a phylogenetic tree. To capture the incomplete sampling, the delayed introduction of imported infections to Switzerland, and potential factors associated with basic reproductive number $R_0$, we extend the branching process model to infer transmission parameters. Overall, the $R_0$ is estimated to be 0.44 (95%-confidence interval 0.42—0.46) and it is decreasing by 11% per 10 years (4%—17%). Our findings indicate rather diminishing HIV transmission among Swiss heterosexuals far below the epidemic threshold. Generally, our approach allows to assess the danger of self-sustained epidemics from any viral sequence data.

DOI: https://doi.org/10.7554/eLife.28721.001

## Introduction

Epidemics of HIV and other blood-borne and sexually transmitted diseases (for instance syphilis, HBV and HCV) can be subdivided into concentrated and generalized epidemics. While for the former, the rapid infectious agent transmission is restricted to core transmission groups involved in high-risk behaviors (such as men who have sex with men and injecting drug users), the generalized

**eLife digest** In epidemiology, the "basic reproductive number" describes how efficiently a disease is transmitted, and represents the average number of new infections that an infected individual causes. If this number is less than one, many people do not infect anybody and hence the transmission chains die out. On the other hand, if the basic reproductive number is larger than one, an infected person infects on average more than one new individual, which leads to the virus or bacteria spreading in a self-sustained way.

Turk et al. have now developed a method to estimate the basic reproductive number using the genetic sequences of the virus or bacteria, and have used it to investigate how efficiently HIV spreads among Swiss heterosexuals. The results show that the basic reproductive number of HIV in this group is far below the critical value of one and that over the last years this number has been decreasing. Furthermore, the basic reproductive number differs for different subtypes of the HIV virus, indicating that the geographical region where the infection was acquired may play a role in transmission. Turk et al. also found that people who are diagnosed later or who often have sex with occasional partners spread the virus more efficiently.

These findings might be helpful for policy makers as they indicate that the risk of self-sustained transmission in this group in Switzerland is small. Furthermore the method allows HIV epidemics to be monitored at high resolution using sequence data, assesses the success of currently implemented preventive measures, and helps to target subgroups who are at higher risk of an infection – for instance, by supporting frequent HIV testing of these people. The method developed by Turk et al. could also prove useful for assessing the danger of other epidemics.

DOI: https://doi.org/10.7554/eLife.28721.002

epidemic refers to fast pathogen spreading in the heterosexual (general) population resulting in higher overall disease prevalence. Mechanistically, the key factor explaining whether the HIV transmission is concentrated or generalized, is the ability of HIV to spread among heterosexuals. If the epidemic in this population is not self-sustained, the HIV epidemic remains concentrated; otherwise the virus is spreading rapidly in the broad population leading to a generalized HIV epidemic.

In most resource-rich settings HIV transmission is concentrated, that is, driven mostly by transmission among men who have sex with men (MSM) and injecting drug users (IDU), whereas the limited transmission among heterosexuals is maintained by either imported infections or spillovers from other transmission groups (*Kouyos et al., 2010*; *von Wyl et al., 2011*; *Ragonnet-Cronin et al., 2016*; *Xiridou et al., 2010*; *Esbjörnsson et al., 2016*; *Sallam et al., 2017*). This suggests that in most Western European countries and similar epidemiological settings the basic reproductive number $R_0$ among heterosexuals is below 1. However, it is not clear how far away from self-sustained the epidemic is in heterosexuals. Moreover, the change in HIV transmission among heterosexuals over time is another important, yet unknown, factor, especially with evidenced increasing risky sexual behavior (*Kouyos et al., 2015*). It is therefore crucial to assess both the transmission and its time trend in order to obtain meaningful insights into the epidemic.

Assessing the subcritical transmission of HIV in the general population shares some methodological similarities with the analysis of stage III zoonoses, for instance, monkeypox (*Wolfe et al., 2007*), which also exhibit stuttering transmission chains. Both cases follow a source-sink dynamics, i.e., a flux of infections from a subpopulation in which the disease is self-sustained to a population where it is not. For the case of stage III zoonoses and tuberculosis, it has been shown that the distribution of outbreak sizes can be used to quantify the pathogen spread (*Blumberg and Lloyd-Smith, 2013b*; *Blumberg and Lloyd-Smith, 2013a*; *Borgdorff et al., 1998*). The fundamental approach of our study is to apply this concept to transmission of HIV in the general population. However, there are two key differences between emerging zoonotic pathogens and human-to-human infectious agents. Firstly, while the contact tracing data are not available for many sexually transmitted infections (STI), the viral sequences carry valuable information about the transmission chain size distribution. Thus, the approach of quantifying transmissibility from chain size distributions needs to be combined with a tool to derive clusters from viral sequences. Compared to the animal-human transmission the delayed introduction of the index case of an STI or blood-borne virus to the subpopulation of

interest plays an important role, especially in viruses like HIV with long infectious periods in the absence of treatment and higher transmissibility during the acute phase (*Marzel et al., 2016*; *Powers et al., 2011*; *Rieder et al., 2010*; *Rodger et al., 2016*; *Hollingsworth et al., 2008*; *Cohen et al., 2011b*; *Cohen et al., 2011a*; *Cohen et al., 2016*). This is especially important because a considerable fraction of HIV cases in heterosexuals is found in migrants (*Del Amo et al., 2004*; *von Wyl et al., 2011*; *European Centre for Disease Prevention and Control/WHO Regional Office for Europe, 2016*). If, for example, a migrant infected with HIV abroad moves to Switzerland in the chronic stage of the infection, he/she has (from the perspective of the Swiss population) lost some transmission potential upon entering Swiss heterosexual transmission network.

In order to quantify the subcritical transmission we combine phylogenetic cluster analysis with an adapted version of a branching process model based estimator that derives the basic reproductive number $R_0$ from the size distribution of transmission chains. We further extend this approach to determine the impact of calendar time and other potential determinants on $R_0$; especially in order to assess whether $R_0$ exhibits an increasing time trend or is high in particular subgroups. Applying this method to the phylogenetic transmission clusters among heterosexuals in the Swiss HIV Cohort Study (SHCS), we can assess transmission of HIV in this population and in particular the risk of a generalized HIV epidemic together with the main determinants of transmission.

## Results

We developed a method to assess how far HIV transmission in populations with basic reproductive number $R_0 < 1$ is from the epidemic threshold, that is, how far it is from being self-sustained in these populations (see Materials and methods). A classical application of this question/method is HIV-1 transmission in heterosexuals in settings with a concentrated epidemic. Heterosexual HIV-1 transmission in Switzerland is a case in point for such a non-self-sustained HIV epidemic. We identified 3,100 transmission clusters among heterosexuals in the SHCS. These clusters were small in size (*Table 1*) and comprised individuals of broad demographic background (see *Table 2*). Based on the most likely geographic origin of the transmission clusters, we classified 1,133 transmission chains as being of Swiss origin, that is, to represent introductions from other transmission groups in Switzerland into the heterosexual population, and 1,967 to be of non-Swiss origin. For these latter transmission chains, we assumed that the $R_0$ of the index case was reduced by a factor of $\rho_{index} = 0.35$ (see Materials and methods). To take into account the imperfect sampling density we fixed the subtype-depending sampling probabilities based on the results from the study by *Shilaih et al. (2016)*, corrected by the proportion of the HIV infected individuals linked to care (80% based on *Kohler et al., 2015*) and the fraction of heterosexuals from the SHCS with an HIV sequence in the phylogenetic tree (57.22%). The model parameters used in this study are summarized in *Table 1* (see Sensitivity analyses, *Appendix 1—figure 1* and *Appendix 1—figure 2* for the corresponding sensitivity analyses).

### $R_0$ of the HIV transmission in Swiss heterosexuals

To obtain an overall estimate for the $R_0$ of HIV transmission in Swiss heterosexuals, the baseline model was fitted to all of the previously described transmission chain data. In this baseline model the $R_0$ was estimated to be 0.44 (95%-confidence interval (CI) 0.42—0.46). The fact that $R_0$ was clearly below 1 (*p*-value <0.001 from one-sided Wald hypothesis testing $H_0 : R_0 = 1$ against the alternative $H_A : R_0 < 1$) indicated that HIV transmission is far away from a self-sustained epidemic.

Although the overall $R_0$ estimate was clearly below 1, individual subtypes represent different epidemiological settings and hence individual subtypes may have $R_0$ closer to the epidemic threshold. The subtype-stratified analyses indeed yielded lower $R_0$ of 0.35 (95%-CI 0.33—0.39) for subtype B as compared to the non-B subtypes (*Figure 1*). The recombinant form CRF02_AG had the highest estimated $R_0$ of 0.62 (95%-CI 0.56—0.69). Despite these differences among the $R_0$ estimates for different subtypes they were all significantly below 1 (with all *p*-values from the one-sided test smaller than 0.001). Therefore, we concluded that there is no danger of a self-sustained HIV epidemic in Swiss heterosexuals of any HIV subtype.

**Table 1.** Transmission chain size distribution and model parameters.

| | Subtype | | | | | | |
| --- | --- | --- | --- | --- | --- | --- | --- |
| | B | C | 01_AE | 02_AG | A | Other | Overall |
| Total number of chains, $n$ (%) | 1643 (53%) | 322 (10%) | 239 (7.7%) | 331 (11%) | 327 (11%) | 238 (7.7%) | 3100 (100%) |
| Chain size, $n$ (%) | | | | | | | |
| 1 | 1437 (87%) | 280 (87%) | 206 (86%) | 272 (82%) | 269 (82%) | 195 (82%) | 2659 (86%) |
| 2 | 158 (9.6%) | 34 (11%) | 31 (13%) | 40 (12%) | 44 (13%) | 36 (15%) | 343 (11%) |
| 3 | 30 (1.8%) | 7 (2.2%) | 1 (0.42%) | 10 (3.0%) | 10 (3.1%) | 6 (2.5%) | 64 (2.1%) |
| 4 | 12 (0.73%) | - | 1 (0.42%) | 6 (1.8%) | 3 (0.92%) | 1 (0.42%) | 23 (0.74%) |
| 5 | 1 (0.06%) | 1 (0.31%) | - | 2 (0.6%) | 1 (0.31%) | - | 5 (0.16%) |
| 6 | 1 (0.06%) | - | - | 1 (0.3%) | - | - | 2 (0.06%) |
| 7 | 1 (0.06%) | - | - | - | - | - | 1 (0.03%) |
| 8 | 2 (0.12%) | - | - | - | - | - | 2 (0.06%) |
| 9 | 1 (0.06%) | - | - | - | - | - | 1 (0.03%) |
| Sampling probability, $p$ (SD) | 0.39 | 0.29 | 0.34 | 0.26 | 0.33 | 0.29 | 0.35 (0.05) |
| Chain origin, $n$ (%) | | | | | | | |
| Swiss ($\rho_{index} = 1$) | 948 (58%) | 36 (11%) | 36 (15%) | 36 (11%) | 47 (14%) | 30 (13%) | 1133 (37%) |
| non-Swiss ($\rho_{index} = 0.35$) | 695 (42%) | 286 (89%) | 203 (85%) | 295 (89%) | 280 (86%) | 208 (87%) | 1967 (63%) |

DOI: https://doi.org/10.7554/eLife.28721.003

## Time trend of the $R_0$

Despite consistently low $R_0$ estimates, an increasing time trend for $R_0$ would impose a potential concern, especially if the time trend would predict a crossing of the epidemic threshold in the near future. To investigate this, we fitted a univariate model with $\log(R_0)$ as a linear function of the establishment date of the transmission chain. We found that overall the $R_0$ is decreasing at a factor 0.89 per 10 years (95%-CI 0.83—0.96). The per subtype-stratified analyses showed the consistently decreasing time trend among the subtypes ranging from factor 0.65 per 10 years for subtype A to 0.89 for B-subtype.

To better capture the changes of $R_0$ over time we included higher-order polynomials of the establishment date to our model (*Figure 2*). With the reference date on the 1st of January 1996 (which corresponds to the median estimated date of infection - see *Table 2*) a cubic spline (without the linear term) was identified as the optimal model according to the Bayesian information criterion (BIC). This model exhibits a mild increase of the $R_0$ from the mid 1980's to the mid 1990's, with a peak-$R_0$ of 0.49 (95%-CI 0.46—0.53) reached in 1996 and followed by a steep and monotonic decrease. It is noteworthy that the time of peak-$R_0$ coincided with the introduction of highly active antiretroviral therapy. Shortly after the $R_0$ started to rapidly decrease and has never rebounded. This extrapolation should be, however, taken with a grain of salt and seen more as a trend rather than a prognosis, since only a few transmission chains have been observed for the recent years (which is reflected by wide confidence intervals).

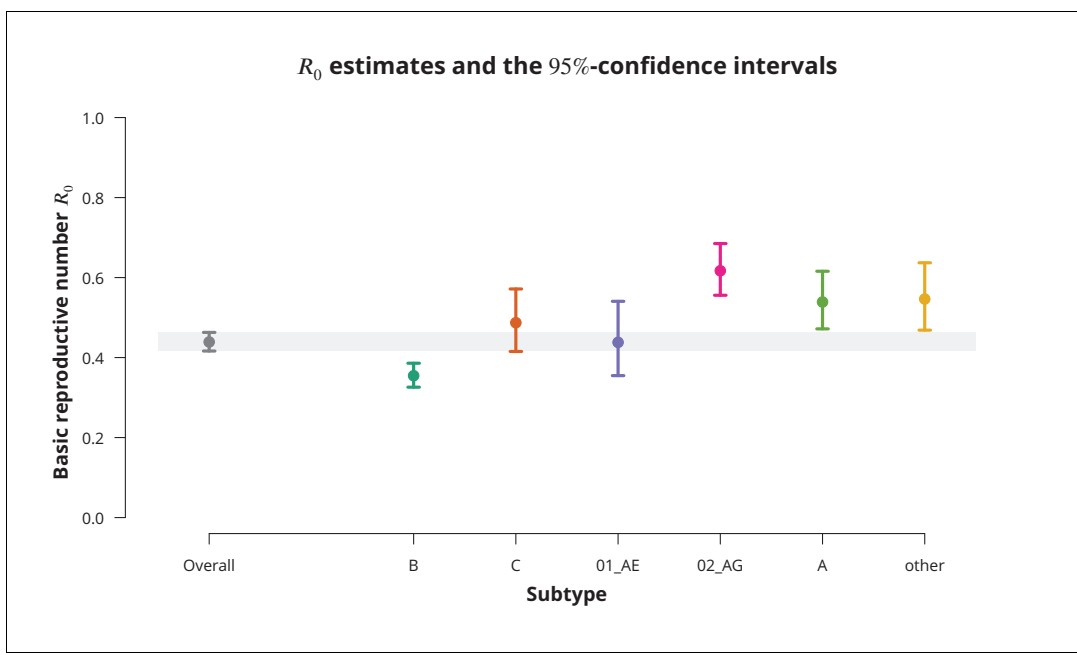

**Figure 1.** Overall basic reproductive number $R_0$ and $R_0$ per subtype from stratified analysis. The dark gray point indicates the overall basic reproductive number $R_0$ estimate (by neglecting the transmission chain subtypes) and the corresponding 95%-confidence interval is shown with the dark gray line and the gray-shaded band. The analogous results from the per-subtype stratified analysis are represented by colored points and lines, each color corresponding to one of the subtypes (B, C, CRF01_AE, CRF02_AG or A) or the group of subtypes (other).

DOI: https://doi.org/10.7554/eLife.28721.004

## Determinants of the HIV-transmission

Finally, we identified the characteristics associated with higher $R_0$ and therefore potential focal sub-populations, in which the basic reproductive number $R_0$ could be above 1. The simplest model containing only the linear terms of risk factors showed that the $R_0$ is decreasing with the establishment date of the transmission chain and that all non-B subtypes have higher $R_0$ compared to subtype B, which was consistent with the findings from the univariate model and per-subtype stratified analyses. Moreover, we found that reporting sex with occasional partners and longer time to HIV diagnosis of the index case are associated with higher $R_0$, whereas the earliest CD4 cell count and the age do not have significant effects (*Figure 3*).

These trends remained robust (*Figure 4*) when allowing the covariables to enter the model non-linearly (for instance as polynomials like in the case of the time trend above). The final multivariate model identified subtype, establishment date of the transmission chain, frequency of reporting sex with occasional partner and time to diagnosis of the index case as the significant risk factors associated with $R_0$ (see Selection of the predictive models). Allowing nonlinear terms for the time to diagnosis provided better goodness-of-fit than the linear model. The steep increase of $R_0$ in the early/acute phase (see *Figure 4*) of the infection indicates the importance of early diagnosis (which is nowadays closely related to early treatment initiation) while the time becomes less relevant in the cases diagnosed late in the chronic phase.

## Discussion

Our approach demonstrates that viral sequences combined with basic demographic information can be successfully used not only to estimate the basic reproductive number $R_0$ of HIV in a subcritical setting and thereby assess the danger of a generalized HIV epidemic but also to shed light on the trends and other determinants of viral transmission. As a proof of concept, this approach was applied to HIV transmission in Swiss heterosexuals, for which we found an $R_0$ far below the epidemic threshold with a decreasing time trend - indicating a low and decreasing danger of a generalized

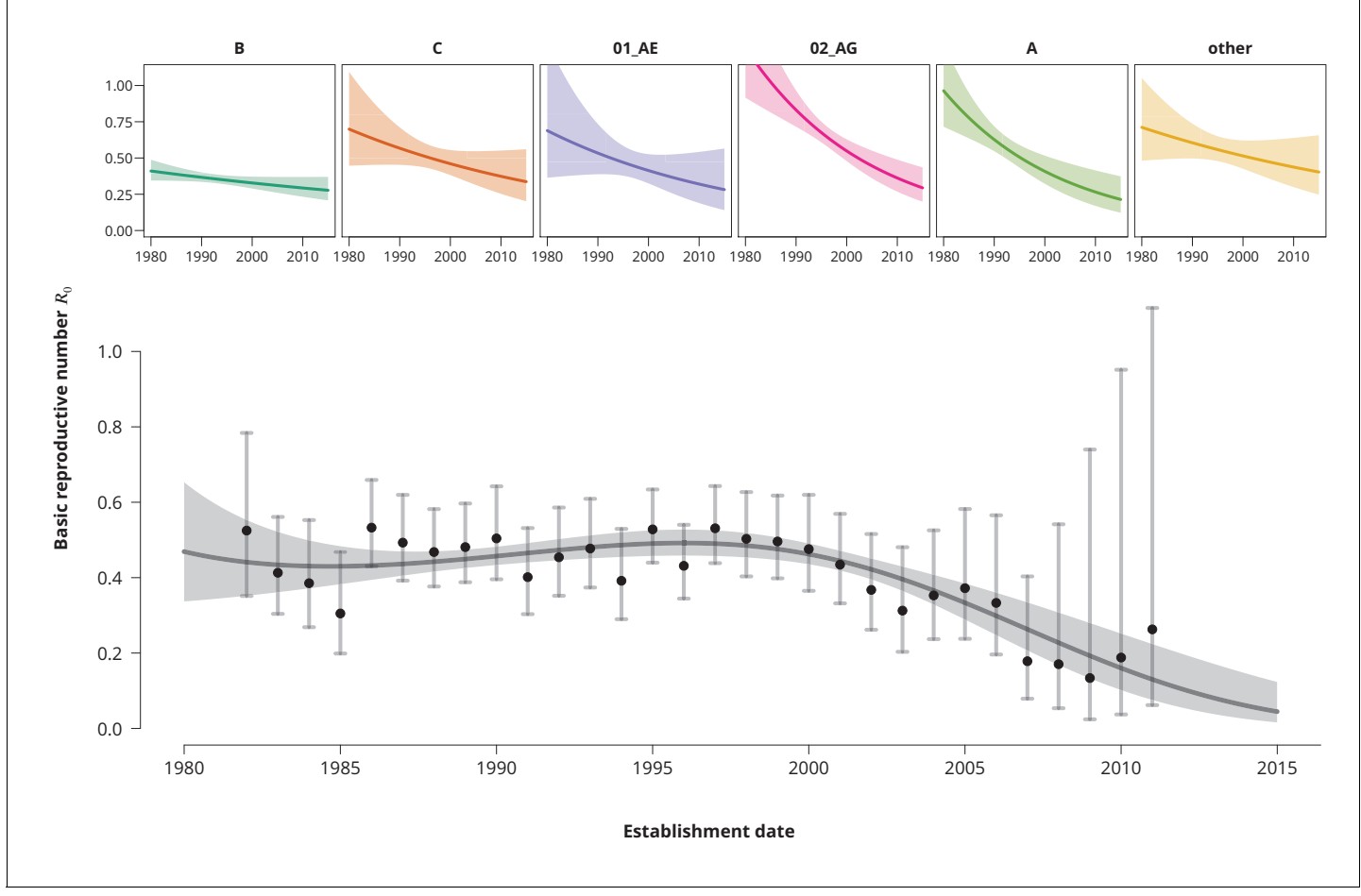

**Figure 2.** Time trends for $R_0$. The upper smaller panels show the time trends for $R_0$ from the subtype-stratified analyses, in which the $log(R_0)$'s were modeled as linear functions of establishment date (i.e., for each subtype the time trend rate was assumed to be constant). The colored shaded-bands correspond to the 95%-prediction bands. The (best-fitting) nonlinear time trend for $R_0$ from the overall analysis is displayed in the lower panel (dark gray curve) together with the 95%-prediction band (gray-shaded area). The black points represent the $R_0$ estimates from the per establishment year stratified analyses and the gray vertical lines the corresponding 95%-confidence intervals.

DOI: https://doi.org/10.7554/eLife.28721.005

**Table 2.** Patients' demographic characteristics.

| | Patients | Transmission chains |
| --- | --- | --- |
| | | Index case |
| Total number, $n$ | 3698 | 3100 |
| Age at estimated date of infection [in years], median (IQR) | 29.2 (23.1—37.8) | 28.8 (22.8—37.4) |
| Estimated date of infection, median (IQR) | Jun 1996 (Sep 1990—Nov 2001) | Nov 1995 (Sep 1989—May 2001) |
| Time to diagnosis [in years], median (IQR) | 3.40 (1.66—5.24) | 3.54 (1.78—5.43) |
| Reported sex with occasional partner [as fraction of FUPs*], median (IQR) | 0.53 (0.09—0.89) | 0.50 (0.07—0.88) |
| No available FUP[†], $n$ (%) | 250 (6.8%) | 226 (7.3%) |
| Earliest CD4 count [per µL][‡], median (IQR) | 310 (143—510) | 300 (134—507) |

*Follow-up visit (FUP).

[†]Patients without FUP questionnaire regarding the sexual risk behavior. See Sensitivity analyses.

[‡]One patient did not have any available CD4 cell count. The missing value was imputed with the mean CD4 cell count.

DOI: https://doi.org/10.7554/eLife.28721.006

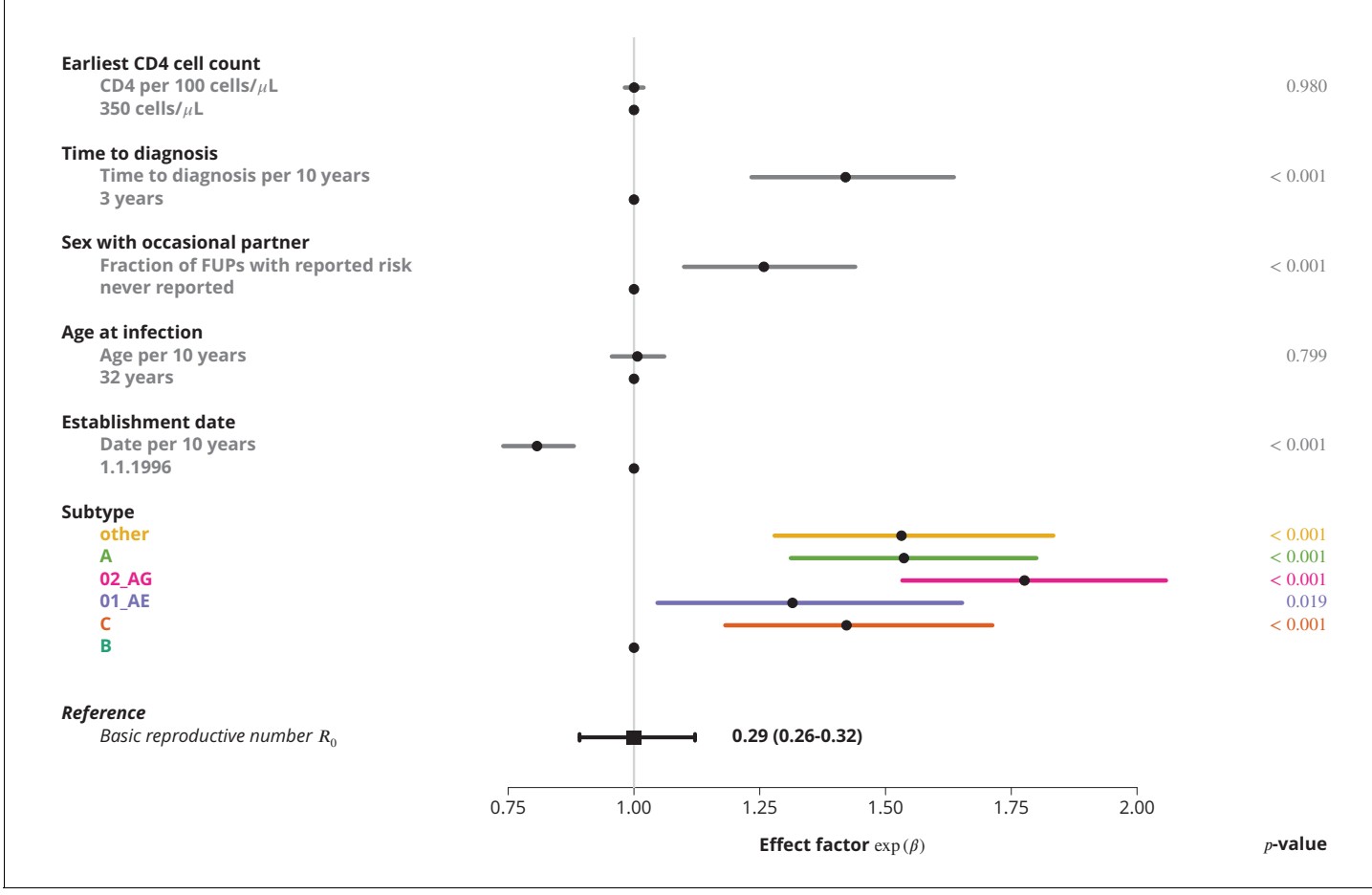

**Figure 3.** Effect of different factors on the basic reproductive number $R_0$ from the multivariate model with only linear factor terms. The black square and the black line show the reference basic reproductive number $R_0$ and its 95%-confidence interval (for a transmission chain of subtype B which started on 1.1.1996, and in which the index case was diagnosed 3 years after the infection, was 32 years old upon infection, never reported on having sex with occasional partner and had the earliest CD4 cell count of 350 cells per μL). The vertical gray line separates the factors associated with lower $R_0$ (left; effect factor <1) and from the factors contributing to higher $R_0$ (right; effect factor >1). The black points on this line refer to the reference transmission chain. The colored and dark gray lines represent the effect sizes from multivariate model (black circles depicting the estimates) for different factors and their 95%-confidence intervals. The corresponding $p$-values are shown in the rightmost column. FUP, follow-up visit.

DOI: https://doi.org/10.7554/eLife.28721.007

epidemic. Even though the Swiss HIV epidemic is captured in outstanding detail and representativeness by the SHCS, our approach can be easily used in other non-self-sustained epidemics since viral sequences from genotypic resistance testing are nowadays routinely produced in most resource-rich settings. Moreover, the generalizability of our approach might be broadened to other settings and viruses due to the increased availability of viral sequences boosted by decreasing sequencing costs and the ability of the method to adjust for imperfect sampling.

To our knowledge our study represents the first systematic assessment of the basic reproductive number for subcritical HIV transmission among heterosexuals, which makes it difficult to compare our results to other estimates. In addition, it was conducted in one of the most densely sampled settings. Most of the studies investigated the transmission route composition of larger transmission clusters across different B and non-B subtypes (*Esbjörnsson et al., 2016*; *Chaillon et al., 2017*; *Ragonnet-Cronin et al., 2016*; *Sallam et al., 2017*; *Kouyos et al., 2010*; *von Wyl et al., 2011*), or focused on homosexual men or injecting drug users as the main drivers of HIV transmission (*Amundsen et al., 2004*). *Stadler et al. (2012)* previously presented a birth-death process based analysis of HIV transmission in Switzerland. However, since this approach is restricted to sufficiently large clusters, it is not suitable for subcritical settings and might potentially overestimate $R_0$ due to

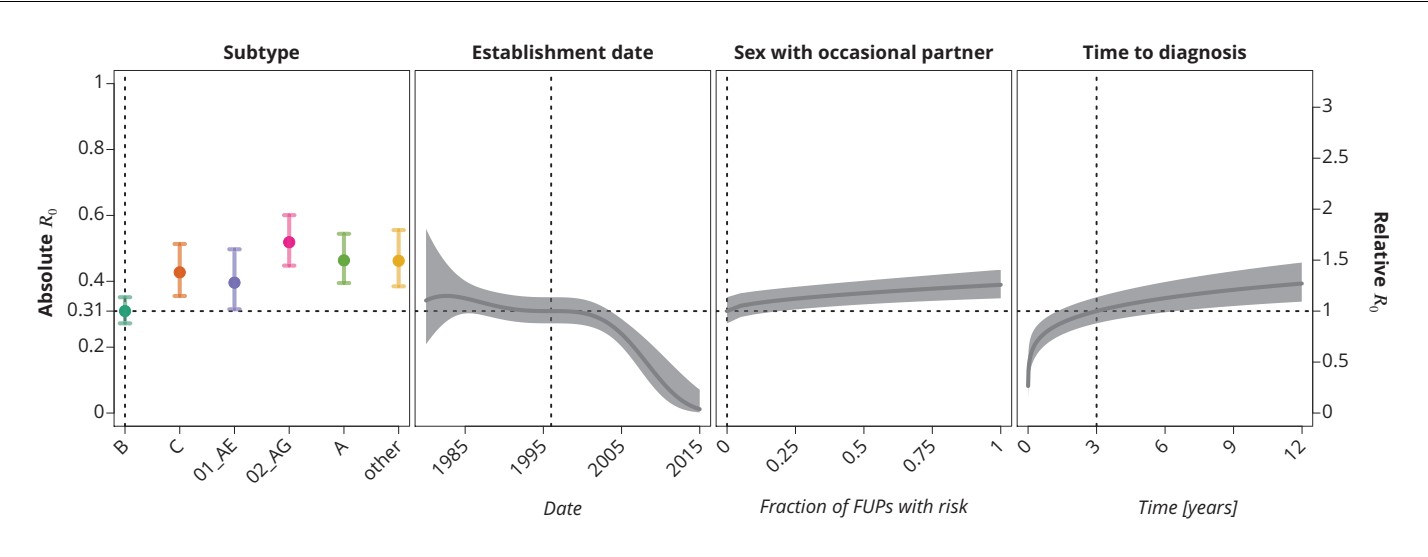

**Figure 4.** Final multivariate model's profile plots of factors associated with the basic reproductive number $R_0$. The vertical dotted lines depict the reference transmission chain (of subtype B, started on 1.1.1996, in which the observed index case did not report having sex with occasional partner and was diagnosed after 3 years after the infection). The left *y*-axis represents the basic reproductive number whereas the right *y*-axis corresponds to the relative values of $R_0$ as compared to the baseline $R_0$. The $R_0$ as the function of specific factor (with the other factors held fixed at the reference value) is displayed by the colored (for HIV-1 subtype) and the dark gray (establishment date, sexual risk behavior and time to diagnosis) lines. The vertical bars and the shaded bands, respectively, correspond to the 95%-confidence intervals.

DOI: https://doi.org/10.7554/eLife.28721.008

selection bias. Hence, our approach, which is tailored to subcritical viral transmission, is complementary to theirs. Among other studies specific for heterosexual populations, *Hughes et al. (2009)* focused on the clusters of size at least 2 across non-B subtypes, and *Xiridou et al. (2010)* studied the impact of sexual behavior of migrants on the HIV prevalence, while none of them directly assessed the danger of self-sustained epidemics.

Epidemiological differences between the HIV-1 subtypes, especially between B and non-B subtypes, have been pointed out previously (*Kouyos et al., 2010*; *von Wyl et al., 2011*). Yet the exact factors contributing to the differences are difficult to identify. On the one hand, the non-B subtypes are often seen in relation to the infections imported from abroad, which could be introduced either by immigrants or by residents who got infected while temporarily abroad. A proportion of these introductions could be attributed to the sex tourism (*Rogstad, 2004*). However, even the differences between the various non-B subtypes could be substantial, as they represent different epidemiological settings. For instance, the CRF01_AE is often found in Asians and it also most likely originates from Southeastern Asia (*Angelis et al., 2015*), while subtypes originating from Africa, such as CRF02_AG (*Mir et al., 2016*), are frequently found in people of black ethnicity. Additionally, poverty and different policies regulating prostitution worldwide also have an impact on the transmission patterns, like on rate of condom use, access to HIV testing and treatment (*Shannon et al., 2015*). On the other hand, disentangling the effect of different epidemiological characteristics and even of the strains remains challenging, as $R_0$ was significantly affected by the HIV subtype even in the multivariate model (*Figure 3*).

One of the key components of our model is the index case relative transmission potential $\rho_{index}$, which is also associated with some degree of uncertainty. To illustrate its role and influence on the transmission parameters we performed a range of sensitivity analyses (*Appendix 1—figure 1*). On the one hand, omitting the reduced transmissibility of the index case, that is, assuming $\rho_{index} = 1$, leads to largely underestimated $R_0$ (overall $R_0$ of 0.33, 95%-CI 0.31—0.35) affirming the importance of this extension. Then again, the concrete value chosen may be debatable, especially due to arguable infectivity in chronic phase (studied by *Bellan et al., 2015*); thus a small $\rho_{index}$ can be caused both by immigration later during chronic infection and by elevated infectivity in the acute phase. To address this issue we lowered the $\rho_{index}$ for the transmission chains of non-Swiss origin to 0.25 to obtain a

more conservative estimate of $R_0$, which was, nevertheless, still safely below 1 (0.47, 95%-CI 0.44—0.49). Furthermore, even though theoretically the transmission potential of some index cases could also be enhanced (i.e., $\rho_{index} > 1$), for instance for sex workers, we do not expect that this is the case for many transmission chains and would therefore have only marginal effect on our estimates. Besides, since a $\rho_{index} > 1$ would lead to even lower $R_0$, our main conclusions would not change (in fact, the assumption of $\rho_{index} < 1$ is conservative with respect to our conclusion of $R_0 < 1$).

The presented model is based on source-sink dynamics, which is reflected in the importance of the index case and its immigration background, while the role of emigration is neglected. However, in many resource-rich settings similar source-sink patterns can be observed, both in the migration related influxes and the new virus introductions in the heterosexual population from other risk groups. Namely, the immigration from a setting with a generalized epidemic to a setting with a concentrated epidemic is by far more likely than the emigration. Similarly, occasional spillovers from other risk groups, such as MSM and IDU, to the generalized population are more probable than the reverse. Therefore, the assumption of absence of such outflow from the epidemiological setting under consideration is not problematic when considering a country like Switzerland, but might present a potential limitation if the unit of interest is smaller, like a region or a city.

Our approach has theoretically several limitations, which we, however, expect to have only moderate impact. First, we assumed stuttering transmission chains, or in other words, that the basic reproductive number $R_0$ is below 1. If $R_0$ was larger than 1 the observed transmission chains would have been much longer (see Sensitivity analyses and *Appendix 1—figure 5*) which is inconsistent with rather small clusters observed in HIV transmission among Swiss heterosexuals (*Kouyos et al., 2010*; *von Wyl et al., 2011* and *Shilaih et al., 2016*). Second, some transmission chains might still be active, meaning that some patients from the chain could be still infectious and therefore able to further spread the virus. The consequence of this would be an underestimation of $R_0$ for recent years. However, given much higher transmissibility of HIV in the acute and recent infection (*Marzel et al., 2016*) and estimated mean time to being non-infectious of approximately 2—2.5 years in recent years (*Stadler et al., 2012*; *Hughes et al., 2009*) the majority of the observed transmission chains had most likely been stopped by the time of sampling and hence we do expect that this issue will not lead to a major bias of our estimates (see Sensitivity analyses and *Appendix 1—figure 4*). Third, since our method is based on transmission clusters their misidentification and negligence of their structure could be another constraint. Possible overlapping transmission chains (as it was also noted in *Blumberg and Lloyd-Smith, 2013b*), that is, misidentifying two transmission chains resulting from two separate introductions of closely related viruses as one single chain, represent the biggest concern in this regard. Failing to identify separate clusters would lead to a higher $R_0$ estimate. However, this means that our method will tend to overestimate $R_0$ and is hence conservative with respect to its main aim of assessing the danger of self-sustained epidemics; thus, if the method predicts an $R_0$ strongly below 1, the corresponding epidemic will indeed be far away from being self-sustained. Moreover, our method neglects the transmission chain structure and consequently uses only the aggregated number of infections, and assumes the same $R_0$ for the entire chain except for the index case. Yet, this issue is likely to have a weak impact, since we focus on subcritical transmission; the transmission chains are hence short (see *Table 1*), and their structure conveys only limited information. Indeed, although a huge variation in sexual behavior has been shown previously (*Liljeros et al., 2001*), our sensitivity analyses exhibited no major impact of varying sexual risk behavior on risk determinants (Sensitivity analyses and *Appendix 1—figure 6*). Finally, even though the negative binomial model was proposed as the favorable choice for the offspring distribution compared to the Poisson distribution (*Blumberg and Lloyd-Smith, 2013b*) we did not observe any significant differences in the $R_0$ estimates (see Sensitivity analyses and *Appendix 1—figure 7*). On the contrary, due to the simplicity of the Poisson distribution we managed to integrate the index case transmission potential reduction and the heterogeneity between the transmission chains into our Poisson-based model in a more systematic manner through the observed variability of the demographic characteristics.

## Conclusion

Generally, our approach allows the assessment of the danger of a concentrated epidemic to become generalized based on the viral sequence data. We demonstrated this approach for the case of heterosexual HIV transmission in Switzerland. In particular, even though the study highlighted some

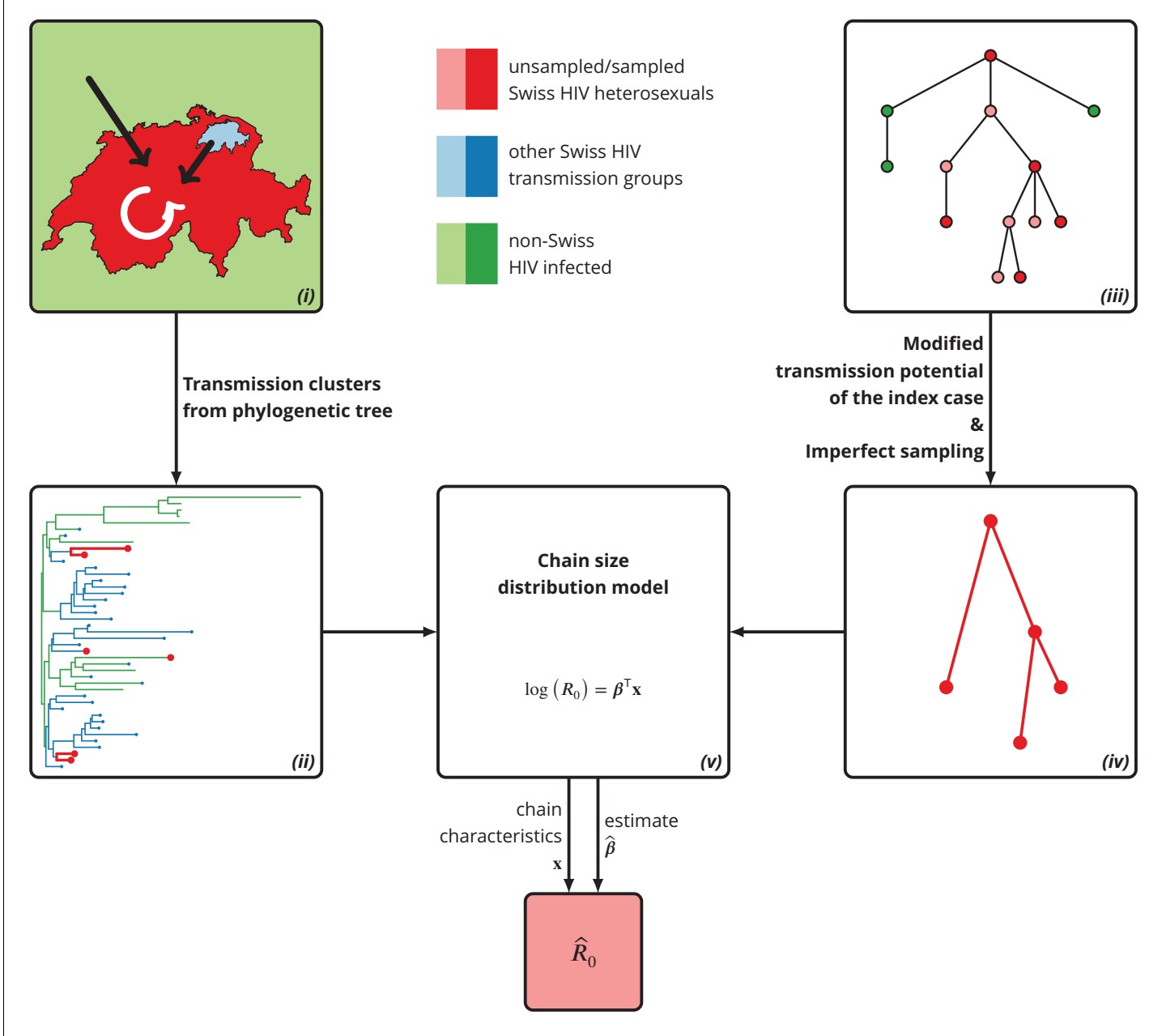

**Figure 5.** Graphical representation of our phylogeny-based statistical approach. (i): HIV transmission among heterosexuals in Switzerland (white arrow) has never led to a self-sustained epidemic. However, the unknown potential of imported infections (black arrows) either from abroad or from other transmission groups in Switzerland remains a large concern. (ii): The HIV transmission chains corresponding to Swiss heterosexuals (depicted in red) were identified from the phylogenetic tree containing the SHCS and background viral sequences. (iii): Our mathematical model is based on the discrete-time branching process with nodes of three different types: sampled Swiss infection (red), unsampled Swiss infection (light red) and foreign infection infected by a Swiss index case before moving to Switzerland (green). (iv): Our method for inferring $R_0$ accounts for both imperfect sampling and modified transmission potential of the index case. (v): Moreover, it includes the baseline transmission chain characteristics to assess the determinants of $R_0$.

DOI: https://doi.org/10.7554/eLife.28721.009

heterogeneity between the HIV subtypes, our findings indicate that there is no imminent danger of a self-sustained epidemic among Swiss heterosexuals, but rather diminishing HIV transmission far below the epidemic threshold. Hence, the HIV epidemic in Switzerland is and most likely will remain restricted to high risk core groups, especially MSM. Moreover, the results suggest that integrated

prevention measures in Switzerland taken over time were successful within the heterosexual population.

## Materials and methods

We combined a phylogenetic cluster detection approach to identify transmission chains in the population under consideration with an adapted version of the model developed in *Blumberg and Lloyd-Smith (2013a)* to infer the basic reproductive number $R_0$ (*Figure 5*). In particular, we accounted for both imperfect detection (included in *Blumberg and Lloyd-Smith, 2013a*) and modified transmissibility of the index case (not included in *Blumberg and Lloyd-Smith, 2013a*) from the perspective of the setting under consideration because it enters the population only (late) in chronic infection – e.g., via immigration. Moreover, we included the baseline transmission chain characteristics (such as HIV-1 subtype, date of infection, time to diagnosis, risky sexual behavior, etc.) to explain the heterogeneity among transmission chains. Note that our approach in principle estimates the effective reproductive number defined as the number of secondary infections for the current state of population; however, in case of a non-self-sustained epidemic with low prevalence, the vast majority of the population is susceptible and hence the effective reproductive number is a very good approximation for the basic reproductive number.

### SHCS and viral sequences

The SHCS is a multicenter, nationwide, prospective observational study of HIV infected individuals in Switzerland, established in 1988 (*Swiss HIV Cohort Study et al., 2010*). The SHCS was approved by the ethics committees of the participating institutions (Kantonale Ethikkommission Bern, Ethikkommission des Kantons St. Gallen, Comite Departemental d'Ethique des Specialites Medicales et de Medicine Communataire et de Premier Recours, Kantonale Ethikkommission Zürich, Repubblica e Cantone Ticino–Comitato Ethico Cantonale, Commission Cantonale d'Étique de la Recherche sur l'Être Humain, Ethikkommission beiderBasel; all approvals are available on http://www.shcs.ch/206-ethic-committee-approval-and-informed-consent), and written informed consent was obtained from all participants. Up to December 2016 over 19,500 patients have been enrolled. The SHCS is highly-representative as it covers more than 75% HIV-positive individuals on antiretroviral therapy (ART) in Switzerland (*Swiss HIV Cohort Study et al., 2010*). In addition to the extensive demographic and clinical data collected at biannual/quarterly follow-up (FUP) visits, for approximately 60% of the patients at least one partial *pol* sequence from the genotypic resistance testing is available (in total 22,036 sequences from the SHCS resistance database until August 2015). The patients with heterosexual contact as the most likely transmission route comprise about one third of all SHCS participants.

### Phylogenetic tree

The phylogenetic tree was constructed from the Swiss HIV sequences of the SHCS patients and non-Swiss background sequences exported from the *Los Alamos National Laboratory, 2016* database (241,783 HIV-1 viral sequences of any subtype and including the circulating recombinant forms 01–74 retrieved on February 23rd, 2016 spanning over the *protease* and *RT* regions with fragments of at least 250 nucleotides; the HXB2 sequence and sequences from Switzerland were removed afterwards). The sequences of 8 HIV-1 subtypes and circulating recombinant forms (B, C, CRF01_AE, CRF02_AG, A(1-2)), G, D and F(1-2)) were pairwise aligned to the reference genome HXB2 (accession number K03455) using Muscle v3.8.31 (*Edgar, 2004*). Sequences with insufficient sequencing quality of the protease region (coverage of less than 200 nucleotides between the positions 2253 and 2549 of HXB2) or reverse transcriptase region (less than 500 nucleotides between positions 2550 and 3869) were excluded. Using the earliest available of the remaining sequences for each patient, the phylogenetic tree was built with the FastTree algorithm under the generalized time-reversible model of nucleotide evolution (*Price et al., 2009*) including 10,840 SHCS and 90,933 background sequences.

### Transmission chains

The Swiss heterosexual transmission chains were defined as clusters in the phylogenetic tree containing exclusively Swiss HIV sequences from individuals with heterosexual contact as the most likely

route of the transmission, regardless of the respective genetic distances and local support values (see Sensitivity analyses and *Appendix 1—figure 8* for alternative definition). The transmission chains and the patients enrolled in the SHCS forming them were identified with custom written functions in R (version 3.3.2).

For each transmission chain we determined if it was introduced to the Swiss HIV heterosexuals either as an imported infection from abroad or from other HIV transmission groups within Switzerland. The geographic origin for a given chain was obtained as the country of the closest sequence, which did not belong to Swiss heterosexuals. Specifically, we considered the smallest clade that contained both the transmission chain and either a non-Swiss or non-heterosexual sequence, and chose the sequence with the smallest pairwise genetic distance to the transmission chain (with respect to the Jukes and Cantor (JC69) model).

Additionally, in each extracted transmission chain the observed index case was identified as the patient with the earliest estimated date of infection in the chain. The date of HIV infection for each single individual was imputed with the model described by *Taffé et al. (2008)* if the patient had enough CD4 cell count measurements before the ART initiation and the estimated date of infection fell within the seroconversion window; otherwise the midpoint of the seroconversion window was used. The demographic characteristics (*Table 2*) of the index case were extracted from the SHCS, including age at infection, time to diagnosis, first available CD4 cell count and sexual risk behavior. The latter was quantified as the fraction of semiannual follow-up visits at which the patient reported sex with occasional partners. The patients with no available questionnaire regarding the sexual risk behavior were assumed to have never reported on having sex with occasional partner (see Sensitivity analyses and *Appendix 1—figure 9* for the corresponding sensitivity analysis). The characteristics of the index case were then used to define the features of each corresponding transmission chain.

## Estimating the basic reproductive number from a model

Our model is based on the basic discrete-time branching process. The basic reproductive number $R_0$ was inferred from the model as the expected number of offsprings, therefore the offspring distribution represents the crucial component of the chain size distribution model. In the following sections we describe the main extensions of the basic branching process theory, which were implemented in our model. The detailed derivations can be found in Appendix 3.

### Offspring distribution

We modeled the offspring distribution in a transmission chain using a Poisson distribution, which is a special case of the negative binomial distribution. The latter has been suggested in the literature (*Blumberg and Lloyd-Smith, 2013b*) in order to infer $R_0$; however since we did not observe any large differences between the two distributions (see Sensitivity analyses and *Appendix 1—figure 7*), we decided to use the simpler Poisson model.

Suppose that $R_{k,n}$ denotes the number of secondary infections of transmission degree $n$ caused by the $k$th individual from the preceding generation (i.e., infected individuals with transmission degree $n-1$), where the transmission degree refers to the number of transmissions needed to transfer the pathogen from the index case (see Appendix 3 for detailed model description). Under the Poisson offspring distribution the number of secondary infections is modeled by

$$R_{k,n} \sim Pois(R_0),$$

which coincides with the definition of the basic reproductive number $R_0 = \mathbb{E}[R_{k,n}]$. Some index cases may have lower transmission potential, e.g., immigrants that arrive during their chronic infection phase, while other index cases may exhibit enhanced transmissibility, for example, sex workers or foreigners living in Switzerland without a partner. To capture a potentially modified transmissibility of the index case we assumed a different offspring distribution of the root, namely

$$R_{1,0} \sim Pois(\rho_{\text{index}} R_0),$$

where $\rho_{\text{index}}$ denotes the index case relative transmission potential.

To assess the trends and determinants of $R_0$, we further extended the offspring distribution based on the baseline characteristics $\mathbf{x}$ of the transmission chain. More precisely, we assumed that the

logarithm of $R_0$ can be linearly described by the chain characteristics which resulted in the offspring distributions

$$R_{k,n} \sim Pois\left(\exp\left(\boldsymbol{\beta}^T \mathbf{x}\right)\right) \quad \text{and} \quad R_{1,0} \sim Pois\left(\rho_{\text{index}} \exp\left(\boldsymbol{\beta}^T \mathbf{x}\right)\right)$$

for the secondary and the index cases, respectively. Hence, the $R_0$ can be predicted from the effect sizes $\boldsymbol{\beta}$ of factors $\mathbf{x}$ as

$$R_0 = \exp\left(\boldsymbol{\beta}^T \mathbf{x}\right).$$

Note that since each transmission chain $i$ has its specific baseline characteristics $\mathbf{x}_i$ (perhaps even sampling density $p_i$ and index case relative transmission potential $\rho_{\text{index},i}$) the notation above represents a simplification. More precisely, the $R_0$ of the $i$th transmission chain equals $R_{0,i} = \exp\left(\boldsymbol{\beta}^T \mathbf{x}_i\right)$.

## Likelihood function

The likelihood function was expressed in terms of the probability generating function (PGF) of the transmission chain size distribution assuming independent and stuttering (i.e., $R_0 < 1$ assures that each transmission chain goes extinct almost surely) transmission chains. The following assumptions were made when incorporating the incomplete sampling of the sequences:

- For each transmission chain at most one observed transmission chain can be extracted from the phylogeny. In other words, all observed cases belonging to the same transmission chain can be identified as the cases forming the corresponding observed transmission chain, although some intermediate transmitters might not have been sampled. For a phylogeny, this represents by a definition a weak assumption; in contrast, for contact tracing approaches missing one ancestor can lead to misidentifying one transmission chain as two or more.
- The sampling density is independent of the transmission chain size or the transmission degree of the individual, namely each case of the transmission chain can be observed independently from the rest of the chain with probability $p$.

Let $T$ denote the true size of a transmission chain and $\widetilde{T}$ the size of the corresponding observed transmission chain. The above two assumptions can be summarized as

$$\widetilde{T} \mid T \sim Bin(T, p),$$

and the PGF $\widetilde{\mathcal{T}}$ of the observed transmission chain size hence equals

$$\widetilde{\mathcal{T}}(z; R_0, \rho_{\text{index}}, p) = \mathcal{T}\left((1-p) + pz; R_0, \rho_{\text{index}}\right)$$

in terms of the PGF $\mathcal{T}$ of $T$. The probability that a transmission chain has observed size of $\tilde{t} \geq 0$ (where $\tilde{t} = 0$ means that none of the cases of the transmission chain is detected) is given by

$$\mathbb{P}\left[\widetilde{T} = \tilde{t}\right] = \frac{1}{\tilde{t}!} \widetilde{\mathcal{T}}^{(\tilde{t})}(0; R_0, \rho_{\text{index}}, p).$$

In particular, the probability that a transmission chain is observed (i.e., the observed size is strictly positive) can be calculated as

$$\mathbb{P}\left[\widetilde{T} > 0\right] = 1 - \mathbb{P}\left[\widetilde{T} = 0\right] = 1 - \widetilde{\mathcal{T}}(0; R_0, \rho_{\text{index}}, p).$$

However, since only the transmission chains with at least one detected case can be extracted from the phylogeny (and therefore to account for the unobserved transmission chains) we are interested in the probability that an observed transmission chain has a specific size. The probability of observing a transmission chain of size $\tilde{t} > 0$ is

$$\mathbb{P}\left[\widetilde{T} = \tilde{t} \mid \widetilde{T} > 0\right] = \frac{\mathbb{P}\left[\widetilde{T} = \tilde{t}\right]}{\mathbb{P}\left[\widetilde{T} > 0\right]} = \frac{1}{\tilde{t}!} \frac{\widetilde{\mathcal{T}}^{(\tilde{t})}(0; R_0, \rho_{\text{index}}, p)}{1 - \widetilde{\mathcal{T}}(0; R_0, \rho_{\text{index}}, p)}.$$

Finally, for a set of independent observed transmission chain sizes $\left\{\tilde{t}_i\right\}_{i=1}^{I}$ the likelihood function equals

$$L\left(R_0 | \{\widetilde{t_i}\}_{i=1}^{I}, \rho_{\text{index}}, p\right) = \prod_{i=1}^{I} \frac{1}{\widetilde{t_i}!} \frac{\widetilde{\mathcal{T}}^{(\widetilde{t_i})}(0; R_0, \rho_{\text{index}}, p)}{1 - \widetilde{\mathcal{T}}(0; R_0, \rho_{\text{index}}, p)}$$

if the same $R_0$, $\rho_{\text{index}}$ and $p$ are assumed for all transmission chains. For transmission chains with different baseline characteristics and different parameters, the generalized likelihood function is

$$L\left(\boldsymbol{\beta} | \{\widetilde{t_i}, \mathbf{x}_i, \rho_{\text{index},i}, p_i\}_{i=1}^{I}\right) = \prod_{i=1}^{I} \frac{1}{\widetilde{t_i}!} \frac{\widetilde{\mathcal{T}}^{(\widetilde{t_i})}\left(0; \exp\left(\boldsymbol{\beta}^T \mathbf{x}_i\right), \rho_{\text{index},i}, p_i\right)}{1 - \widetilde{\mathcal{T}}\left(0; \exp\left(\boldsymbol{\beta}^T \mathbf{x}_i\right), \rho_{\text{index},i}, p_i\right)}.$$

## Model fit

The maximum likelihood (ML) estimator for $\boldsymbol{\beta}$, the predictor for $R_0$ and the corresponding statistics (confidence intervals, $p$-values, etc.) were implemented in the R package *PoisTransCh* (*Turk, 2017*, https://github.com/tejaturk/PoisTransCh; copy archived at https://github.com/elifesciences-publications/PoisTransCh). The provided confidence intervals are the Wald-type 95%-confidence intervals (see Sensitivity analyses for the comparison against different types) and the $p$-values are based on the Wald statistic. Initially, we assessed the impact of covariables potentially associated with HIV transmission. Specifically, we considered HIV-1 subtype, establishment date of the transmission chain (i.e., the earliest estimated date of infection in the transmission chain), reported sex with occasional partner, age at infection, first measured CD4 cell count and time to diagnosis of the index case. Final model selection was carried out by the forward selection and backward elimination algorithms based on the Akaike and Bayesian information criterion (AIC and BIC, respectively). The detailed steps are provided in Selection of the predictive models.

## Datasets

Previously published datasets from *Kouyos et al. (2010)* and *von Wyl et al. (2011)* were used in this study. As previously discussed in these publications, due to the large sampling density this data would, in principle, allow for the reconstruction of entire transmission networks and could thereby endanger the privacy of the patients. This is especially problematic because HIV-1 sequences frequently have been used in court cases. Therefore, a random subset of 10% of the sequences are accessible via GenBank. These accession numbers are as follows: GU344102-GU344671, EF449787, EF449788, EF449796, EF449798, EF449828, EF449829, EF449838, EF449844, EF449852, EF449853, EF449854, EF449860, EF449880, EF449883, EF449889, EF449895, EF449901, EF449904, EF449905, EF449917, EF449921, EF449928, EF449930, EF449943, EF449950, EF449960, EF449971, EF449980, EF449987, EF450004, EF450005, EF450011, EF450024, EF450026, GQ848113, GQ848120, GQ848140, GQ848145, GQ848149, JF769777-JF769851

## Acknowledgements

We thank Mohaned Shilaih for providing the original data regarding the sampling density. Furthermore, we thank Alex Marzel, Katharina Kusejko, Bruno Ledergerber and Roland R Regoes for fruitful discussions. We thank the patients who participate in the Swiss HIV Cohort Study (SHCS); the physicians and study nurses for excellent patient care; the resistance laboratories for high-quality genotypic drug resistance testing; SmartGene (Zug, Switzerland) for technical support; Johannes Abegglen, Bojana Milosevic, Alexandra U Scherrer, Anna Traytel, and Susanne Wild from the SHCS Data Center (Zurich, Switzerland) for data management; and Danièle Perraudin and Mirjam Minichiello for administrative assistance.

## Additional information

### Group author details

**Swiss HIV Cohort Study**

V Aubert; M Battegay; E Bernasconi; J Böni; DL Braun; HC Bucher; A Calmy; M Cavassini; A Ciuffi; G Dollenmaier; M Egger; L Elzi; J Fehr; J Fellay; H Furrer; CA Fux; HF Günthard; D Haerry; B Hasse;

HH Hirsch; M Hoffmann; I Hösli; C Kahlert; L Kaiser; O Keiser; T Klimkait; RD Kouyos; H Kovari; B Ledergerber; G Martinetti; B Martinez de Tejada; C Marzolini; KJ Metzner; N Müller; D Nicca; G Pantaleo; P Paioni; A Rauch; C Rudin; AU Scherrer; P Schmid; R Speck; M Stöckle; P Tarr; A Trkola; P Vernazza; G Wandeler; R Weber; S Yerly

## Competing interests

Enos Bernasconi: E.B. has been a consultant for BMS, Gilead, ViiV Healthcare, Pfizer, MSD, and Janssen; has received unrestricted research grants from Gilead, Abbott, Roche, and MSD; and has received travel grants from BMS, Boehringer Ingelheim, Gilead, MSD, and Janssen. Hansjakob Furrer: The institution of H.F. has received unrestricted grant support from ViiV, Gilead, Abbott, Janssen, Roche, Bristol-Myers Squibb (BMS), Merck Sharp & Dohme (MSD), and Boehringer Ingelheim. Huldrych F Günthard: H.F.G. has been an adviser and/or consultant for GlaxoSmithKline, Abbott, Gilead, Merck, Novartis, Boehringer Ingelheim, Roche, Tibotec, Pfizer, and BMS and has received unrestricted research and educational grants from Roche, Abbott, BMS, Gilead, Astra-Zeneca, GlaxoSmithKline, and MSD (all money to the institution). Roger D Kouyos: R.D.K. has received speaker honoraria and travel grants from Gilead Sciences. None if these are in relation with the submitted manuscript. The other authors declare that no competing interests exist.

## Funding

| Funder | Grant reference number | Author |
|---|---|---|
| Schweizerischer Nationalfonds zur Förderung der Wissenschaftlichen Forschung | 33CS30-148522 | Huldrych F Günthard |
| Yvonne-Jacob Foundation | | Huldrych F Günthard |
| University of Zurich's Clinical Research Priority Program's ZPHI | | Huldrych F Günthard |
| Schweizerischer Nationalfonds zur Förderung der Wissenschaftlichen Forschung | 159868 | Huldrych F Günthard |
| Schweizerischer Nationalfonds zur Förderung der Wissenschaftlichen Forschung | PZ00P3-142411 | Roger D Kouyos |
| Schweizerischer Nationalfonds zur Förderung der Wissenschaftlichen Forschung | BSSGI0-155851 | Roger D Kouyos |

The funders had no role in study design, data collection and interpretation, or the decision to submit the work for publication.

## Author contributions

Teja Turk, Conceptualization, Software, Formal analysis, Visualization, Methodology, Writing—original draft, Writing—review and editing; Nadine Bachmann, Software, Formal analysis, Methodology, Writing—review and editing; Claus Kadelka, Formal analysis, Methodology, Writing—review and editing; Jürg Böni, Sabine Yerly, Vincent Aubert, Thomas Klimkait, Manuel Battegay, Enos Bernasconi, Alexandra Calmy, Matthias Cavassini, Hansjakob Furrer, Matthias Hoffmann, Investigation, Writing—review and editing; Huldrych F Günthard, Conceptualization, Supervision, Funding acquisition, Investigation, Writing—original draft, Writing—review and editing; Roger D Kouyos, Conceptualization, Formal analysis, Supervision, Funding acquisition, Methodology, Writing—original draft, Writing—review and editing

## Author ORCIDs

Teja Turk, http://orcid.org/0000-0003-3065-8578
Nadine Bachmann, https://orcid.org/0000-0002-7303-9542
Hansjakob Furrer, https://orcid.org/0000-0002-1375-3146
Roger D Kouyos, http://orcid.org/0000-0002-9220-8348

## Ethics

Human subjects: The SHCS was approved by the ethics committees of the participating institutions (Kantonale Ethikkommission Bern, Ethikkommission des Kantons St. Gallen, Comite Departemental d'Ethique des Specialites Medicales et de Medicine Communataire et de Premier Recours, Kantonale Ethikkommission Zürich, Repubblica e Cantone Ticino-Comitato Ethico Cantonale, Commission Cantonale d'Étique de la Recherche sur l'Être Humain, Ethikkommission beiderBasel; all approvals are available on http://www.shcs.ch/206-ethic-committee-approval-and-informed-consent), and written informed consent was obtained from all participants.

## Decision letter and Author response

Decision letter https://doi.org/10.7554/eLife.28721.028
Author response https://doi.org/10.7554/eLife.28721.029

## Additional files

### Supplementary files
• Transparent reporting form
DOI: https://doi.org/10.7554/eLife.28721.010

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

## Appendix 1

DOI: https://doi.org/10.7554/eLife.28721.011

## Sensitivity analyses

### Relative transmission potential of the index case

To assess the role of the index case relative transmission potential we carried out three different sensitivity analyses regarding parameter $\rho_{index}$. Firstly, we varied the $\rho_{index}$ for the transmission chains of non-Swiss origin from 0.05 to 1.5. Secondly, we assumed the same $\rho_{index}$ for all transmission chains regardless of their origin and fit the models over a range of $\rho_{index}$ values. Finally, we restricted the analysis only to the transmission chains of non-Swiss origin and varied $\rho_{index}$.

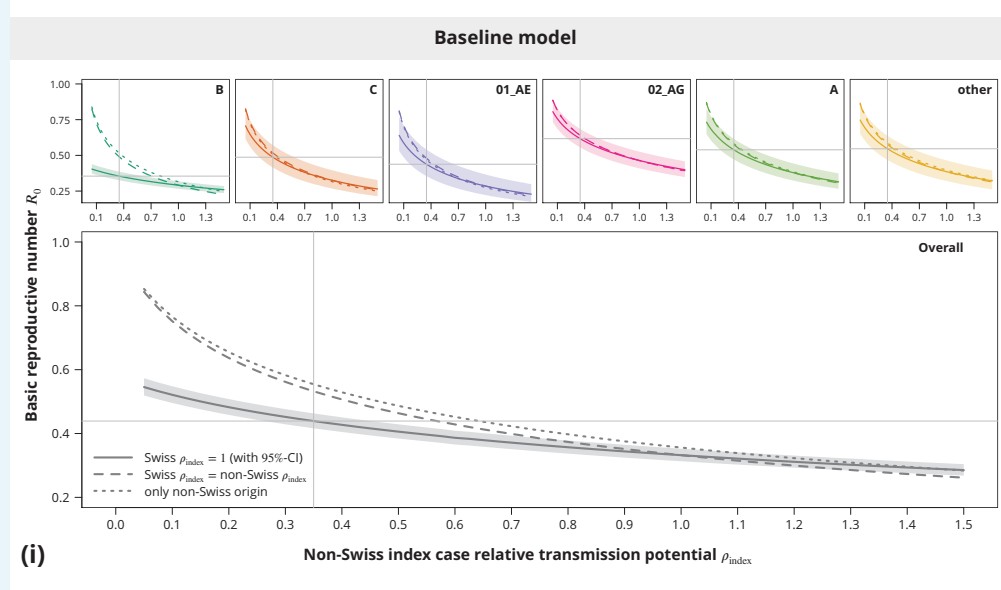

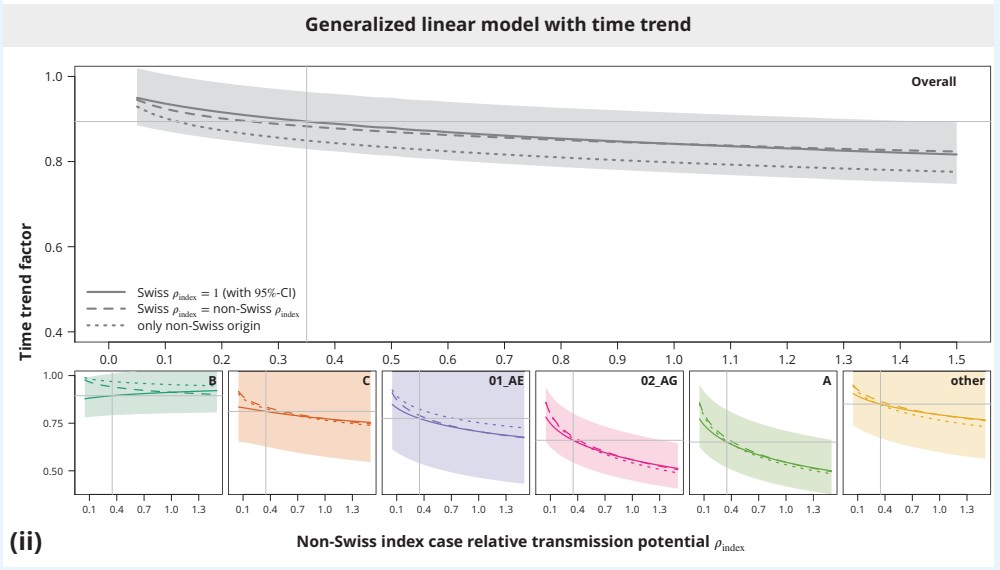

**Appendix 1—figure 1.** Sensitivity analysis regarding the index case relative transmission potential. Panel (i) shows the sensitivity of the $R_0$ estimates from baseline model and panel (ii) the sensitivity of the time trend factor. The colored lines represent the subtype-stratified analyses, while the results from the overall models are shown in gray. In the first sensitivity analysis, the

$\rho_{\text{index}}$ of Swiss-originating transmission chains was held at 1 and the $\rho_{\text{index}}$ of non-Swiss origin varied (solid lines). In the second analysis, the $\rho_{\text{index}}$ of Swiss and non-Swiss origin was the same (dashed lines). The dotted lines show the results from the sensitivity subanalysis including only the transmission chains of non-Swiss origin. The vertical and horizontal lines depict the parameters and estimates from the main analysis, respectively.

DOI: https://doi.org/10.7554/eLife.28721.012

These sensitivity analyses (see *Appendix 1—figure 1*) implied that the conclusion of no danger for a self-sustained epidemic is stable with respect to $\rho_{\text{index}}$ even in the case when some of the Swiss-originated transmission chains are misclassified. In addition, while slightly higher $R_0$ estimates in the non-Swiss transmission chain subanalysis were mostly driven by the non-B subtypes, the results were safely below 1 indicating the non-sensitivity of the main conclusion also when some non-Swiss transmission chains would be falsely identified as such.

## Sampling density

To study the impact of the sampling densities we performed subtype-stratified sensitivity analyses as well as the overall sensitivity analysis by keeping the sampling density constant among the transmission chains. In all scenarios, we varied the sampling density between 0.02 and 1, while $\rho_{\text{index}}$ remained the same as in the main analyses.

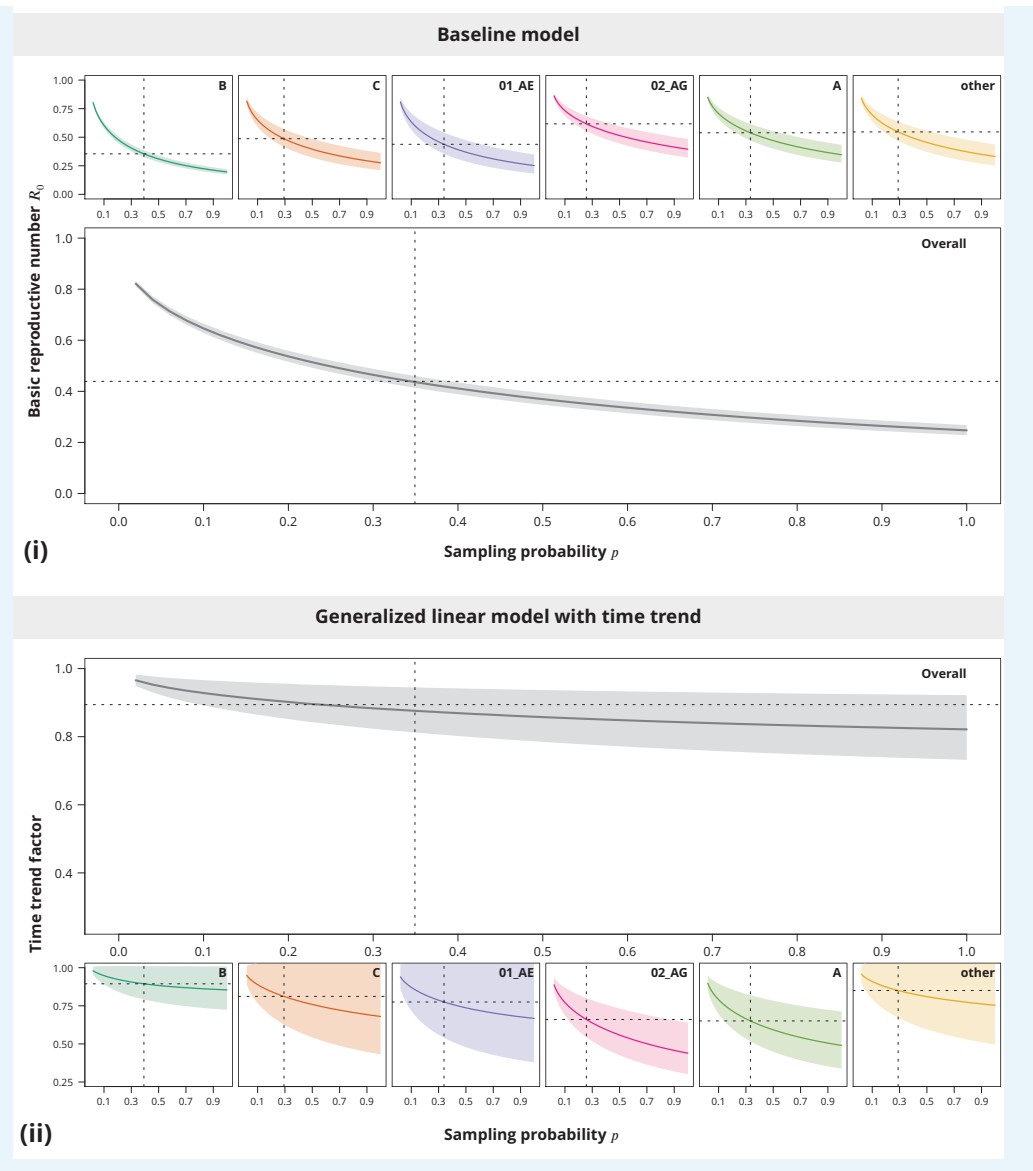

**Appendix 1—figure 2.** Sensitivity analysis regarding the sampling density. The index case relative transmission potential parameter $\rho_{\text{index}}$ was the same as used in the main analyses, while the sampling densities varied ($x$-axis). In the pooled analysis (larger plots) the sampling density was the same for all transmission chains. Panel (i) shows the corresponding estimates of the basic reproductive number $R_0$ and the time trend factor estimates are displayed in panel (ii). The dotted vertical lines depict the sampling densities used for each subtype in our study (subtype-stratified plots) and the mean sampling density over all transmission chains (overall plots). The horizontal dotted lines represent the estimates from the main analysis.

DOI: https://doi.org/10.7554/eLife.28721.013

The sensitivity analyses (see **Appendix 1—figure 2**) showed that neither the $R_0$ from the baseline model nor the time trend are sensitive to the sampling density, namely the conclusions of $R_0$ being significantly below 1 and decreasing time trend could be made even for slightly lower or higher sampling densities.

## Ongoing transmission and stuttering transmission chains assumption

### Duration of infectious period in relation to ongoing transmission

Some of the observed transmission chains may still experience ongoing transmission due to either not yet diagnosed cases or unsuppressed patients within the transmission chain who still have the ability to spread the virus. The transmission chain sizes might thus be too small and $R_0$ underestimated. However, the gradually increasing treatment success (*Castilla et al., 2005*; *Kohler et al., 2015*), benefits of earlier ART initiation (*Kitahata et al., 2009*; *INSIGHT START Study Group et al., 2015*) and consequently updated treatment guidelines (*Günthard et al., 2016*) resulted in a shorter duration of infectious period. Transmission chains which started earlier are thus more strongly affected by ongoing transmission than recent transmission clusters.

One possibility to assess this issue is to investigate the highest possible transmission degree that has completed a transmission at a given time point; that is the maximum number of generations which are not infectious anymore and therefore have used their transmission potential. We assumed that the length of the infectious period is changing linearly with calendar year and fitted a linear regression model to the duration of infectious period of the index cases (measured by time to suppression or treatment start). To ensure a more conservative approach we truncated the fitted infectious period durations from below, such that the minimum was 3 years. Let $d(\tau)$ define the infectious period duration of an individual who became infected at time $\tau$. The worst-case scenario in the context of ongoing transmission and related potential bias is represented by a transmission chain, in which each infected individual transmits the virus just at the end of his/her infectious period. The (conservative) maximum number of completed transmission degrees at time $\tau$ of a transmission chain $i$ that started at $\tau_0$ therefore equals

$$N_{\max,i}(\tau) = \max\{k \in \mathbb{N} | \tau_k \leq \tau\},$$

where $\tau_k$ denotes the latest possible time at which the transmission of the $k$th generation was complete and is calculated iteratively as $\tau_{k+1} := \tau_k + d(\tau_k)$ for $k \in \mathbb{N}_0$ (*Appendix 1—figure 3*). If its index case is still infectious at time $\tau$, it can still produce new infections (which would have a transmission degree 1) and hence $N_{\max,i} = 0$.

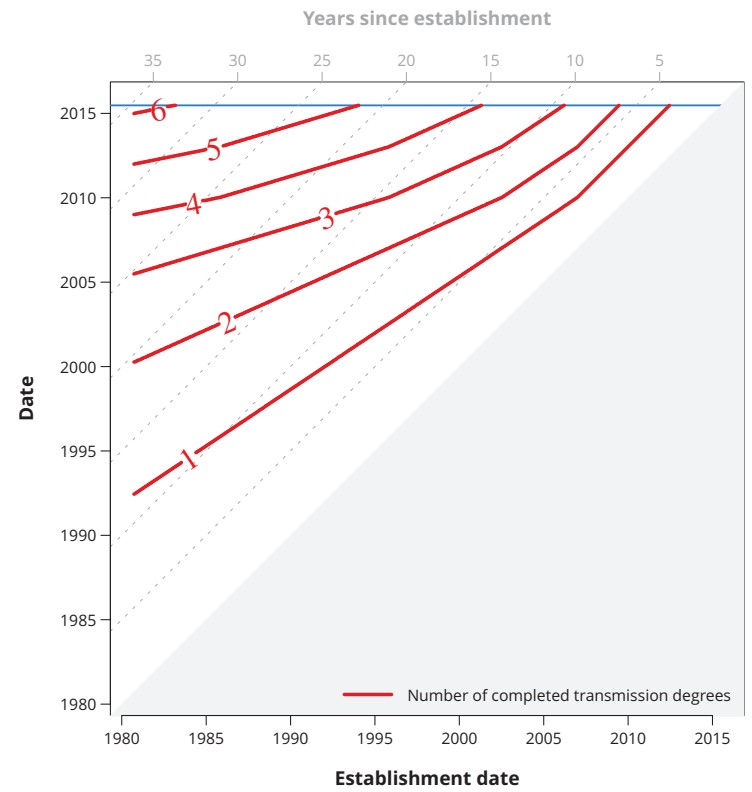

**Appendix 1—figure 3.** Conservative (with respect to ongoing transmission) maximum number of completed transmission degrees by a given date. The red lines show the date ($y$-axis) by which at least a certain number (red numbers) of transmission degrees have been completed for a transmission chain with a specific establishment date ($x$-axis). The diagonal dotted gray lines depict the number of years since the establishment date, and the horizontal blue line represents the last sampling date.

DOI: https://doi.org/10.7554/eLife.28721.014

## Ongoing transmission

To assess the potential bias due to ongoing transmission we compared the estimates based on the transmission chains formed by the cases with the estimated date of infection before a specific date ($\widehat{\omega}$) and based on the transmission chains that had been completed (with respect to the last sampling date) by the same date ($\omega$). The relative bias arising from neglecting the ongoing transmission hence equals

$$\delta_{\text{rel}} = \frac{\widehat{\omega} - \omega}{\omega}.$$

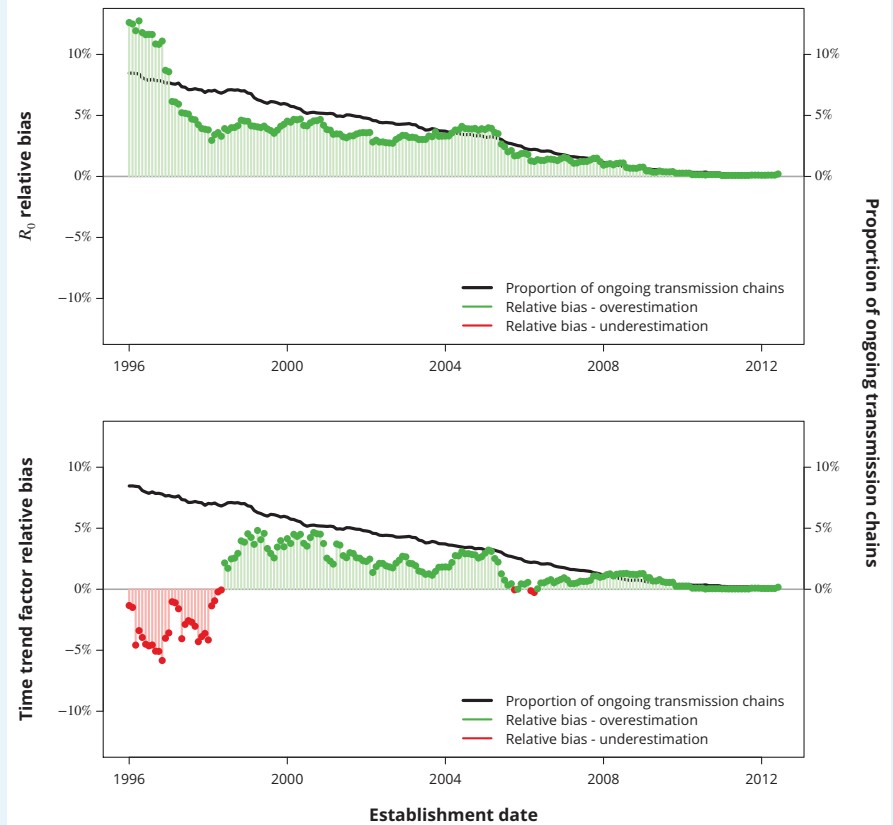

**Appendix 1—figure 4.** Relative bias due to ongoing transmission. The upper panel shows the relative bias of the basic reproductive number $R_0$ from the baseline model and the lower panel the relative bias of the linear time trend factor from the corresponding generalized linear model. The proportion of active transmission chains over time is represented by the black line. The relative bias associated with overestimation and underestimation is displayed with green and red bars-points, respectively. Absence of bias is depicted by the horizontal gray lines.

DOI: https://doi.org/10.7554/eLife.28721.015

The proportion of ongoing transmission chains is decreasing with time, which is in line with a decreasing duration of infectious period, hence indicating that the ongoing transmission is less of an issue for recent years than for older transmission chains. Our sensitivity analyses revealed that the expected bias stemming from neglecting the ongoing transmission is less than 5% since the early 2000's for both key questions (*Appendix 1—figure 4*): the basic reproductive number $R_0$ and its linear time trend factor. Moreover, the relative bias is positive for most of the recent dates, implying that the negligence of ongoing transmission results in rather conservative estimates with respect to our conclusions.

## Subcritical transmission assumption

Like the models described by Blumberg and Lloyd-Smith (*Blumberg and Lloyd-Smith, 2013b*; *Blumberg and Lloyd-Smith, 2013a*), our model also implicitly assumes subcritical transmission. To justify that the extracted HIV transmission chain sizes of the Swiss heterosexuals did not violate this assumption, we simulated transmission chains for various $R_0$ (including the estimated $R_0$) and compared the empirical quantiles between the simulated transmission chain sizes and the transmission chain sizes extracted from the phylogenetic tree. Since some transmission chains (observed or simulated) might still exhibit ongoing transmission at the time of the sampling, we restricted the maximal number of generations (i.e., transmission degrees), which were simulated according to the duration of infectious periods (*Appendix 1—figure 3*).

More precisely, from each observed Swiss heterosexual transmission chain we kept sampling transmission chains (for different 'known true' $R_0$ scenarios) with the maximal number of

simulated generations until a simulated transmission chain was observed (i.e., at least one case was observed) to reflect the more realistic observed transmission chain size distribution. We repeated these steps for each extracted Swiss heterosexual transmission chain.

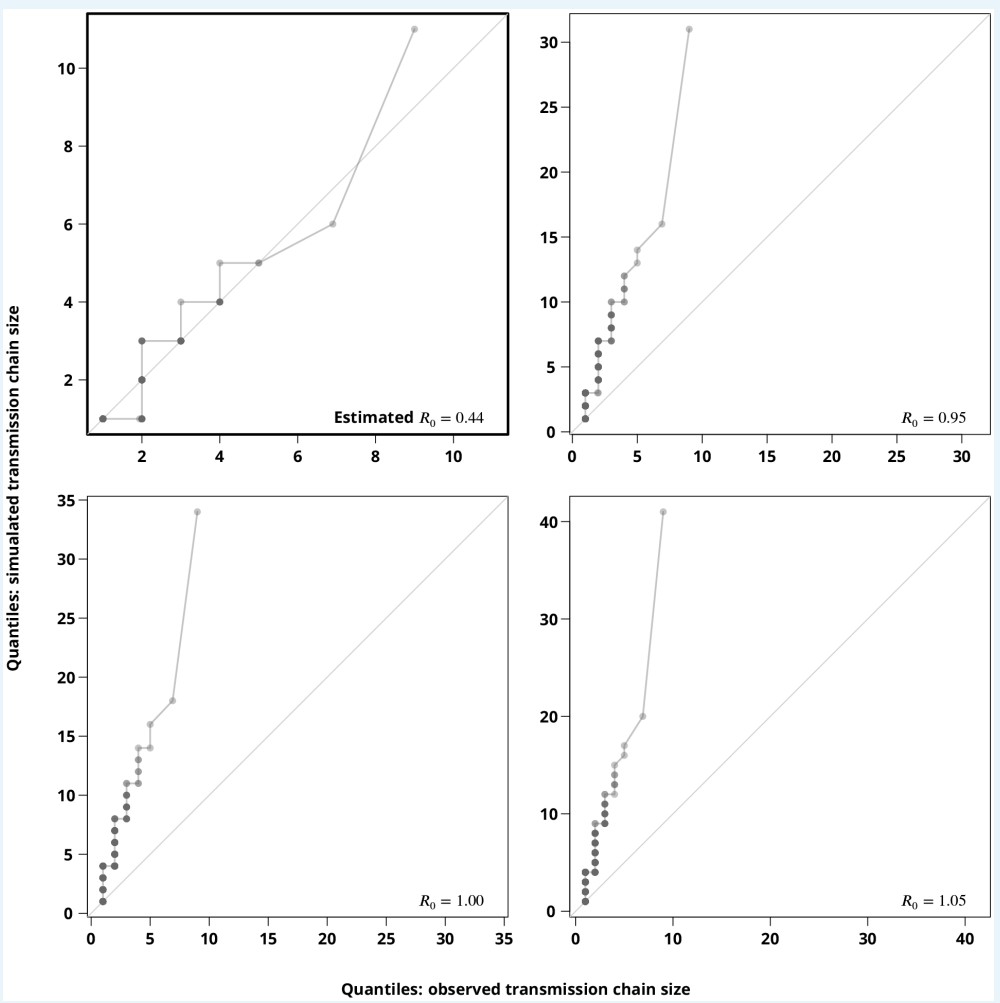

**Appendix 1—figure 5.** Sensitivity analysis regarding the stuttering transmission chains assumption. The Q-Q plots compare the hypothetical transmission chain size distributions ($y$-axis showing their empirical permilles) with the transmission chain size distribution (empirical permilles on the $x$-axis) inferred from the phylogeny. The upper left plot compares the distribution of the simulated transmission chain sizes based on the estimated $R_0$ with the (from the phylogeny) observed transmission chain sizes and thus verifies the $R_0$ estimate. The remaining plots compare the simulated transmission chain size distributions against the extracted transmission chain sizes for $R_0$ closer to 1 to justify the subcritical transmission assumption. Each point represents a permille, hence the darker points indicate more overlapping permilles.

DOI: https://doi.org/10.7554/eLife.28721.016

Finally, we compared the 1000-quantiles (permilles) of the transmission clusters extracted from the phylogeny against simulated transmission chains (**Appendix 1—figure 5**). The Q-Q plots clearly show that the extracted transmission chains would be indeed much longer (the largest observed transmission chain would be of size greater than 30) if the true $R_0$ were above 1 (or even close to 1). Moreover, the size distribution of the transmission chains simulated for the estimated $R_0$ showed a good concordance with the observed transmission chains (upper left Q-Q plot of **Appendix 1—figure 5**).

## Variation in sexual behavior along transmission chains

Our model assumes constant sexual risk behavior along transmission chains. In this sensitivity analysis we assessed how a changing sexual risk behavior would affect our conclusions. We approached this question by slightly changing the definition of the sexual risk behavior of each transmission chain, while the other characteristics stayed the same.

- Instead of the index case determining the risk behavior for each transmission chain a randomly sampled infected individual from the transmission chain was chosen to determine the sexual risk behavior of the transmission chain. Noteworthy, this only affects the minority of the transmission chains, namely those with the observed length $\geq 2$. The multivariate model including only the linear terms was then fitted to the transmission chains with slightly modified sexual risk behaviors. We repeated this 1000 times to get the empirical distribution of the effect sizes on $R_0$ (**Appendix 1—figure 6**).
- We considered the reported sex with an occasional partner on the level of a transmission chain as a proxy for its sexual risk behavior. More precisely, we used the fraction of FUPs of all infected individuals in a transmission chain in which any of these patients reported sex with occasional partner. We then fitted the same multivariate model with only linear terms as in the main analysis and compared the effect sizes and directions (**Appendix 1—figure 6**).

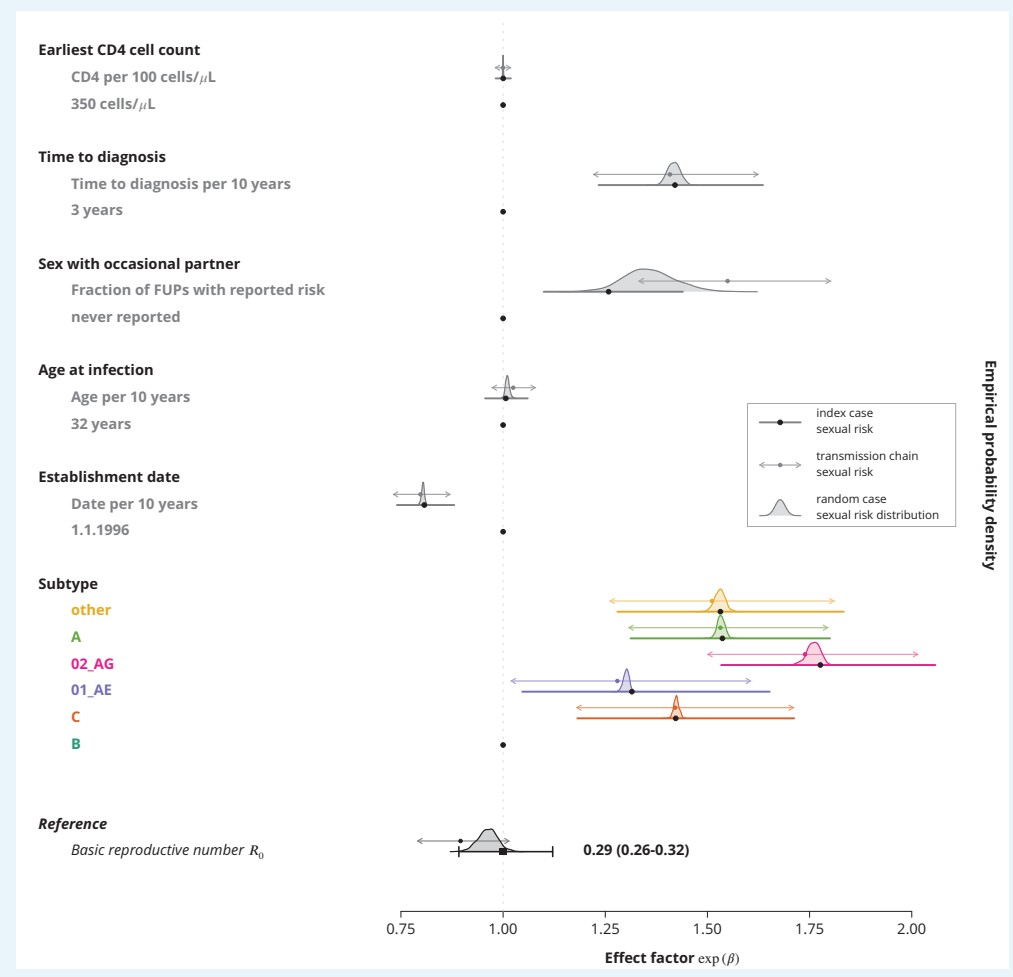

**Appendix 1—figure 6.** Comparison of effect sizes in the multivariate model with linear terms only for different sexual risk behavior definitions of a transmission chain. The thick lines with black circles show the original effect sizes (where the index case determined the sexual risk behavior of the transmission chain) and their 95%-confidence intervals. The empirical distribution of the effect sizes where a random individual in a transmission chain determines its

sexual risk behavior is displayed by the shaded areas. The thinner horizontal double sided arrows with the filled circles correspond to the effect sizes and their 95%-confidence intervals for the transmission chain level fraction of follow-up visits (FUPs) with reported sex with occasional partner by any of the infected individuals from the transmission chain. The vertical dotted gray line depicts the reference $R_0$ from the original model, i.e., using the index case to define the sexual risk behavior.
DOI: https://doi.org/10.7554/eLife.28721.017

Our transmission chains are short in size; therefore we did not expect to see a huge impact of the variations in sexual behavior on the effects. Indeed, the analyses revealed that even with the modified definitions of the risky sexual behavior (and therefore addressing its variation) the effect directions did not change, while the effects sizes did not exhibit a huge difference. In particular, the significance of all risk determinants at the 5% level remained the same.
These findings indicate that the simplification of the equal distribution for the number of secondary infections does not exhibit a dramatic impact on the outcomes in the case of short transmission chains, which dominate in subcritical settings.

## Comparison between Poisson and negative binomial offspring distribution based models

To evaluate the rationale of using the simpler Poisson model we compared the estimates from the baseline models over a range of sampling densities for both Poisson and negative binomial offspring distribution. Since an implementation with modified transmission potential of the index case is not available for the negative binomial model, we conducted the sensitivity analyses with a fixed $\rho_{\mathrm{index}} = 1$.

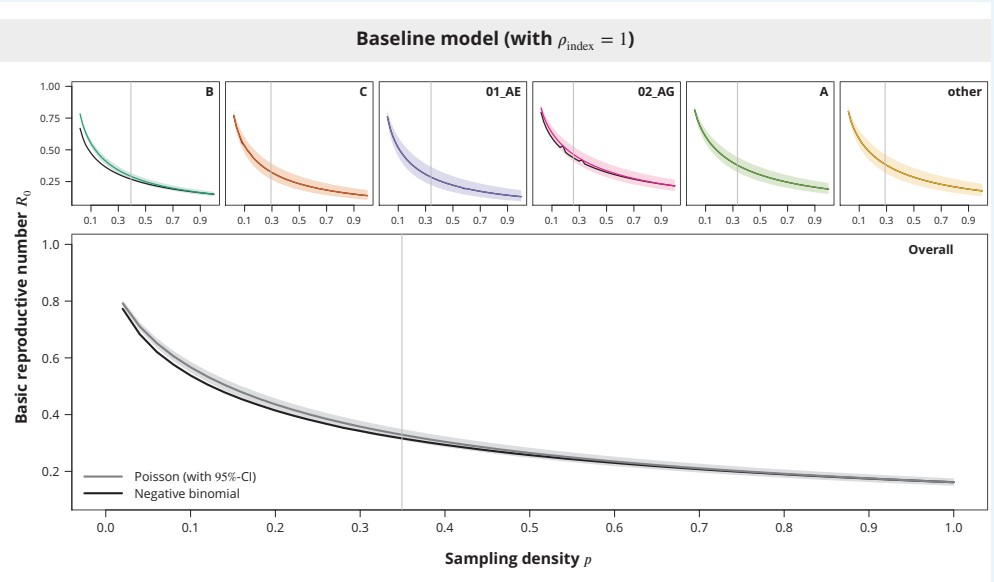

**Appendix 1—figure 7.** Comparison between the Poisson and the negative binomial offspring distribution baseline model $R_0$ estimates. The dark gray and colored lines show the estimates from the model with Poisson offspring distribution, while the black lines correspond to the negative binomial distribution. The index case relative transmission potential parameter $\rho_{\mathrm{index}}$ was fixed to 1 and the sampling density (x-axis) varied. In the overall analysis the sampling density was the same for all transmission chains regardless of their subtype. The vertical gray lines depict the sampling densities used for each subtype in our study (above panels) and the mean sampling density in the overall analysis (bottom panel).
DOI: https://doi.org/10.7554/eLife.28721.018

While the $R_0$ estimates for the majority of the non-B subtypes were practically equal between the two models (see **Appendix 1—figure 7**), the observed differences in the overall analysis

and in the case of B and 02_AG subtypes were mostly larger for low sampling densities. However, we also found that the Poisson model provided rather conservative $R_0$ estimates and therefore this should not affect our main conclusions.

In addition, we performed a likelihood ratio test to evaluate if the multivariate linear negative binomial model (with $\rho_{index} = 1$) is significantly better than the corresponding Poisson model (from **Figure 3**). The $p$-value of $0.74$ indicated no strong preference of the negative binomial over the Poisson model. Noteworthy, this implies that modelling the variability among the transmission chains in terms of their characteristics sufficiently explains the heterogeneity (dispersion parameter $\xi$ of the negative binomial distribution) between the infected heterosexuals forming these transmission chains.

## Relaxed transmission cluster definition

We defined the Swiss heterosexual transmission chains as clusters on the phylogeny containing $100\%$ viral sequences belonging to Swiss heterosexuals. To assess the impact of this definition we relaxed the $100\%$ threshold to $75\%$. All the sequences belonging to the Swiss heterosexuals from these clusters formed more liberally defined transmission chains.

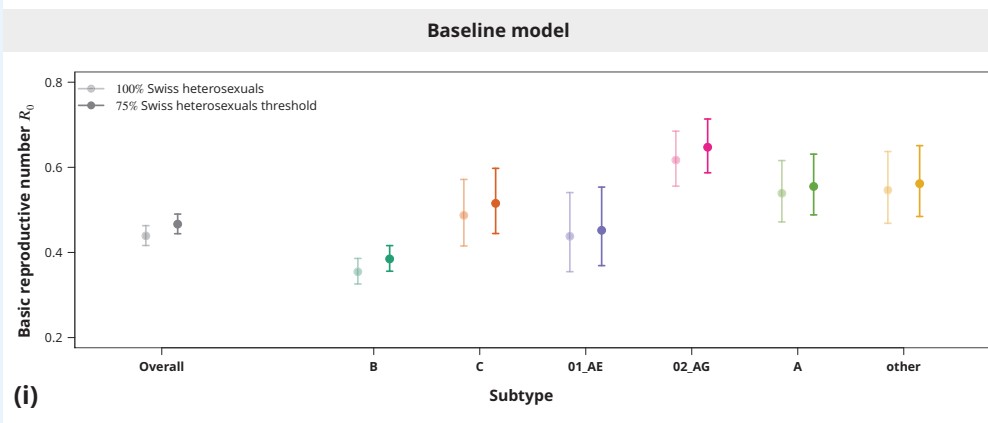

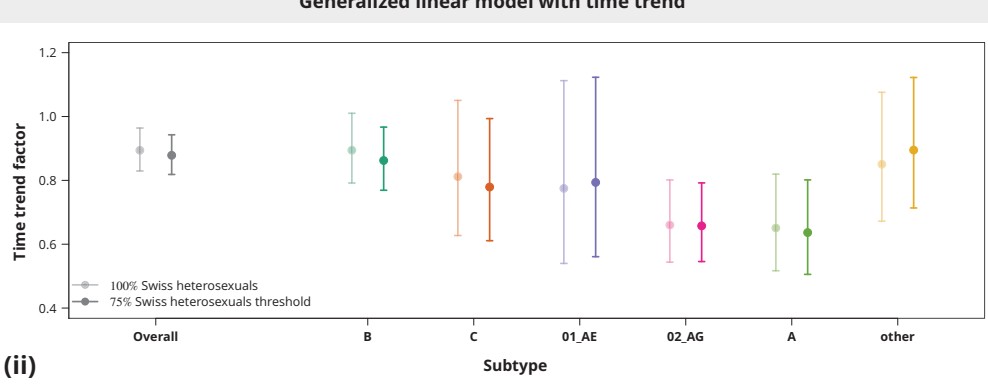

**Appendix 1—figure 8.** Sensitivity analysis regarding the transmission cluster definition. The upper panel (i) compares the estimated $R_0$ with the original cluster definition (brighter lines) with the $R_0$ estimated based on the relaxed cluster definition (darker lines) from the overall analysis (in gray) and subtype-stratified analyses (in colors). Similarly, the bottom panel (ii) shows the comparison between the estimated time trend factors obtained from the transmission chain sizes based on different cluster definition thresholds.

DOI: https://doi.org/10.7554/eLife.28721.019

With the relaxed threshold, we identified 3,039 transmission chains and repeated the main analyses (**Appendix 1—figure 8**). As expected the $R_0$ slightly increased, but stayed below 1. Overall, we did not observe any noteworthy deviations.

## Missing follow-up data for reported sex with occasional partner

In the main analysis of the possible determinants of HIV transmission we imputed missing follow-up information regarding sex with occasional partner with never reporting it (which is equivalent to 0 reporting rate). To evaluate this imputation, we fitted the same multivariate model with linear terms to the subset of the transmission chains in which the data about the sex with occasional partner of the index case was available. However, the effect sizes did not change dramatically; in particular, the effect directions did not change and the same set of determinants was found to be significant (*Appendix 1—figure 9*).

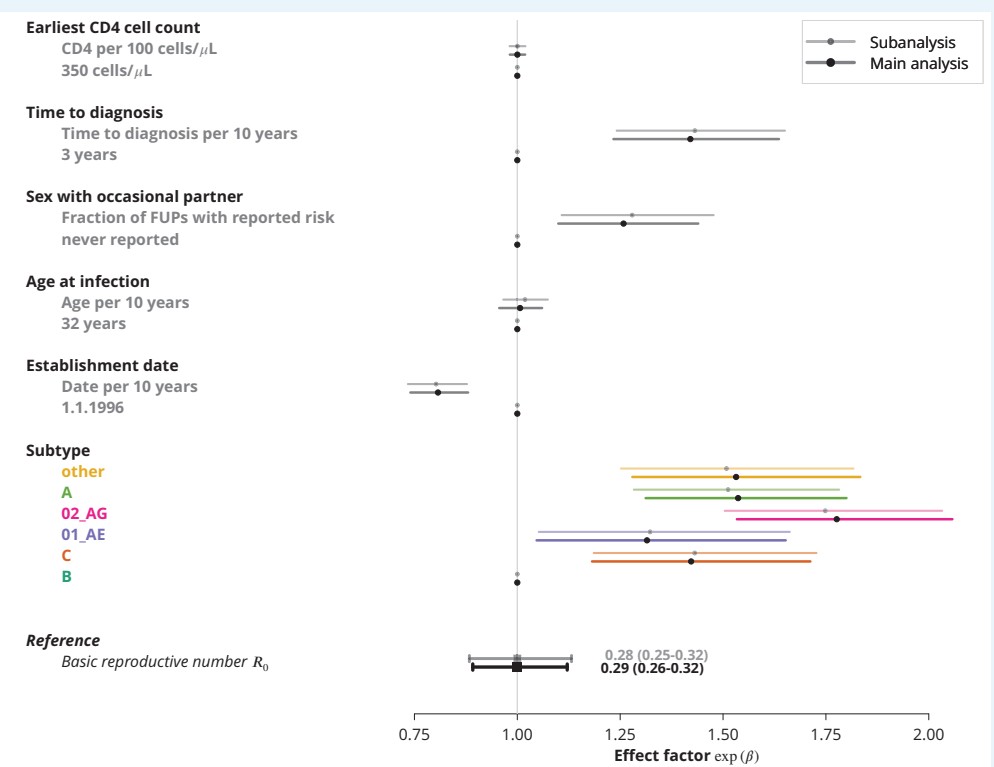

**Appendix 1—figure 9.** Subanalysis for the transmission chains with available follow-up information about sex with occasional partner of the index case compared to the main analysis with imputed data. The effect sizes from the subanalysis are shown in brighter colors and those from the main analysis in dark. In the main analysis, the missing data were replaced by never reporting sex with an occasional partner.

DOI: https://doi.org/10.7554/eLife.28721.020

## Confidence intervals

In our study we used the normal approximation of the ML estimator to construct the 95%-CIs and the prediction intervals. To verify the reliability of this assumption we considered bootstrap and profile likelihood based CIs for each of the models.

For the parametric bootstrap, we sampled $B = 1000$ new datasets of transmission chains from the estimated transmission parameters (i.e., under the assumption that our estimated parameters are the true parameters) for each model. To ensure that the newly sampled datasets had the same sample size, in each repetition $b \in \{1, \dots, B\}$ we kept simulating from each transmission chain until the new transmission chain had at least one observed infection (i. e., such that the observed length was positive). Finally, for each sampled dataset we fitted the same model, extracted the estimated transmission parameters and the corresponding Wald-

type 95%-CIs. The overview of the parameters and the models is provided in *Appendix 1—table 1*.

**Appendix 1—table 1.** Overview of all the parameters, their estimates and the 95%-confidence intervals fitted in all the models presented in this study.

| Subtypes | Parameter number | Parameter name | Parameter estimate | Wald-type 95%-CI | Profile likelihood 95%-CI |
|---|---|---|---|---|---|
| Overall | 1 | $\log(R_0)$ | −0.823 | (−0.876, −0.770) | (−0.878, −0.772) |
| B | 2 | $\log(R_0)$ | −1.037 | (−1.121, −0.952) | (−1.124, −0.955) |
| C | 3 | $\log(R_0)$ | −0.719 | (−0.879, −0.559) | (−0.892, −0.571) |
| 01_AE | 4 | $\log(R_0)$ | −0.826 | (−1.036, −0.615) | (−1.057, −0.632) |
| 02_AG | 5 | $\log(R_0)$ | −0.483 | (−0.587, −0.378) | (−0.594, −0.384) |
| A | 6 | $\log(R_0)$ | −0.618 | (−0.751, −0.485) | (−0.760, −0.492) |
| other | 7 | $\log(R_0)$ | −0.605 | (−0.758, −0.451) | (−0.771, −0.461) |
| Overall | 8 | $\log(R_{0,ref})$ | −0.839 | (−0.894, −0.784) | (−0.895, −0.785) |
|  | 9 | $\frac{Date_{infection}-1.1.1996}{365\cdot10}$ | −0.112 | (−0.187, −0.037) | (−0.188, −0.037) |
| B | 10 | $\log(R_{0,ref})$ | −1.070 | (−1.165, −0.975) | (−1.169, −0.979) |
|  | 11 | $\frac{Date_{infection}-1.1.1996}{365\cdot10}$ | −0.112 | (−0.234, 0.010) | (−0.236, 0.008) |
| C | 12 | $\log(R_{0,ref})$ | −0.692 | (−0.851, −0.533) | (−0.864, −0.544) |
|  | 13 | $\frac{Date_{infection}-1.1.1996}{365\cdot10}$ | −0.209 | (−0.466, 0.049) | (−0.473, 0.046) |
| 01_AE | 14 | $\log(R_{0,ref})$ | −0.781 | (−0.991, −0.570) | (−1.013, −0.588) |
|  | 15 | $\frac{Date_{infection}-1.1.1996}{365\cdot10}$ | −0.255 | (−0.616, 0.106) | (−0.629, 0.101) |
| 02_AG | 16 | $\log(R_{0,ref})$ | −0.434 | (−0.539, −0.329) | (−0.545, −0.333) |
|  | 17 | $\frac{Date_{infection}-1.1.1996}{365\cdot10}$ | −0.415 | (−0.609, −0.222) | (−0.615, −0.226) |
| A | 18 | $\log(R_{0,ref})$ | −0.725 | (−0.892, −0.558) | (−0.907, −0.571) |
|  | 19 | $\frac{Date_{infection}-1.1.1996}{365\cdot10}$ | −0.430 | (−0.660, −0.199) | (−0.672, −0.209) |
| other | 20 | $\log(R_{0,ref})$ | −0.600 | (−0.754, −0.446) | (−0.767, −0.456) |
|  | 21 | $\frac{Date_{infection}-1.1.1996}{365\cdot10}$ | −0.162 | (−0.397, 0.073) | (−0.403, 0.072) |
| Overall | 22 | $\log(R_{0,ref})$ | −0.710 | (−0.780, −0.640) | (−0.782, −0.641) |
|  | 23 | $\left(\frac{Date_{infection}-1.1.1996}{365\cdot10}\right)^2$ | −0.313 | (−0.451, −0.176) | (−0.457, −0.182) |
|  | 24 | $\left(\frac{Date_{infection}-1.1.1996}{365\cdot10}\right)^3$ | −0.184 | (−0.283, −0.086) | (−0.288, −0.091) |
| Overall | 25 | $\log(R_{0,ref})$ | −1.252 | (−1.366, −1.137) | (−1.369, −1.140) |
|  | 26 | $Subtype_C$ | 0.352 | (0.167, 0.538) | (0.158, 0.531) |
|  | 27 | $Subtype_{01\_AE}$ | 0.274 | (0.046, 0.502) | (0.029, 0.490) |
|  | 28 | $Subtype_{02\_AG}$ | 0.575 | (0.428, 0.721) | (0.426, 0.720) |
|  | 29 | $Subtype_A$ | 0.430 | (0.271, 0.588) | (0.266, 0.584) |
|  | 30 | $Subtype_{other}$ | 0.426 | (0.247, 0.606) | (0.238, 0.600) |
|  | 31 | $\frac{Date_{infection}-1.1.1996}{365\cdot10}$ | −0.214 | (−0.301, −0.127) | (−0.301, −0.128) |
|  | 32 | $\frac{Age-32}{10}$ | 0.007 | (−0.045, 0.058) | (−0.046, 0.057) |
|  | 33 | $\frac{CD4-350}{100}$ | 0.000 | (−0.018, 0.019) | (−0.019, 0.018) |
|  | 34 | $Rate_{risk}$ | 0.230 | (0.095, 0.364) | (0.096, 0.365) |
|  | 35 | $\frac{Years_{diagnosis}-3}{10}$ | 0.351 | (0.210, 0.492) | (0.207, 0.490) |

*Appendix 1—table 1 continued*

| Subtypes | Parameter number | Parameter name | Parameter estimate | Wald-type 95%-CI | Profile likelihood 95%-CI |
|---|---|---|---|---|---|
| | 36 | $\log\left(R_{0,ref}\right)$ | $-1.173$ | $(-1.301, -1.045)$ | $(-1.304, -1.048)$ |
| | 37 | $\frac{1}{10}\log\left(\frac{Years_{diagnosis}}{3}\right)$ | $1.727$ | $(1.049, 2.405)$ | $(1.064, 2.420)$ |
| | 38 | $Subtype_C$ | $0.322$ | $(0.140, 0.505)$ | $(0.131, 0.498)$ |
| | 39 | $Subtype_{01\_AE}$ | $0.246$ | $(0.020, 0.472)$ | $(0.004, 0.460)$ |
| | 40 | $Subtype_{02\_AG}$ | $0.516$ | $(0.374, 0.659)$ | $(0.372, 0.658)$ |
| Overall | 41 | $Subtype_A$ | $0.404$ | $(0.246, 0.562)$ | $(0.241, 0.558)$ |
| | 42 | $Subtype_{other}$ | $0.401$ | $(0.223, 0.580)$ | $(0.214, 0.574)$ |
| | 43 | $\left(\frac{Date_{infection} - 1.1.1996}{365 \cdot 10}\right)^3$ | $-0.231$ | $(-0.337, -0.124)$ | $(-0.345, -0.131)$ |
| | 44 | $\sqrt{Rate_{risk}}$ | $0.230$ | $(0.094, 0.366)$ | $(0.096, 0.368)$ |
| | 45 | $\left(\frac{Date_{infection} - 1.1.1996}{365 \cdot 10}\right)^4$ | $-0.129$ | $(-0.227, -0.031)$ | $(-0.235, -0.038)$ |

DOI: https://doi.org/10.7554/eLife.28721.021

For a single parameter $\beta$ (under the assumption that the true value equals the estimated value $\widehat{\beta}$) we therefore obtained a sample of ML estimators $\widehat{\beta}^{(1)}, \ldots, \widehat{\beta}^{(B)}$, from which we estimated the kernel densities and compared them to the normal approximation densities used in the Wald CIs construction (*Appendix 1—figure 10*). Moreover, from the sample of Wald-type 95%-CIs we calculated the coverage rate as the proportion of these CIs that contained the true value $\widehat{\beta}$.

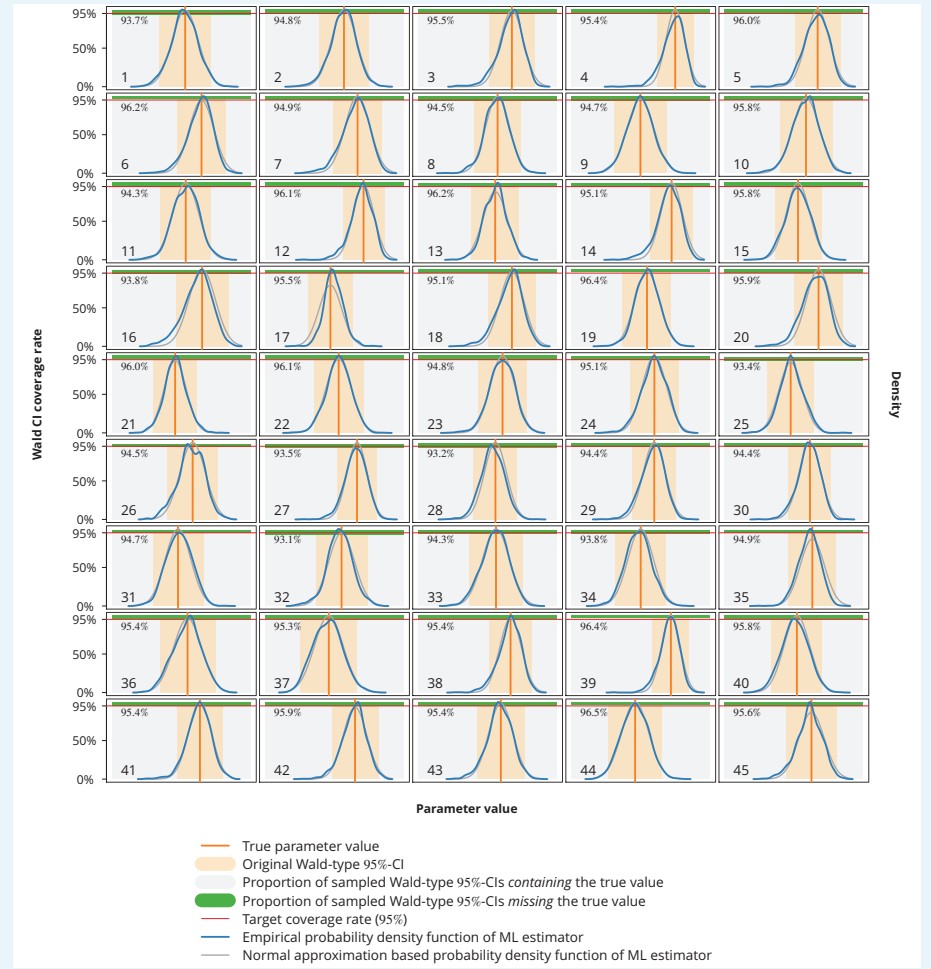

**Appendix 1—figure 10.** Empirical distribution of maximum likelihood (ML) estimator and the Wald-type confidence intervals (CI) coverage rates. Each plot represents a single parameter from a single model (see *Appendix 1—table 1* for the parameters overview including their values), where the number in the lower left corner denotes the parameter's consecutive parameter number. The light gray-shaded area represents the proportion of the Wald-type 95%-CIs from the parametric bootstrap simulations which contained the true value (depicted by the vertical orange line), while the green-shaded area corresponds to those CIs from the simulations that missed the true value. The numbers in the upper left corners are the coverage rates from the parametric bootstrap. The original Wald 95%-CIs used in our study are displayed with the light orange-area. The dark blue and gray lines show the empirical distribution of ML estimators from the parametric bootstrap samples and the normal approximation based probability density function, respectively. The horizontal red lines depict the target coverage rate of 95%.

DOI: https://doi.org/10.7554/eLife.28721.022

Comparing the empirical distribution of the ML estimator from these simulations (*Appendix 1—figure 10*) with the normal approximation from the Wald test, we concluded that the latter represents a valid approximation. In addition, the coverage rates were all very close to the target 95% or above.

Next, in addition to the parametric bootstrap as described above, we also performed a nonparametric bootstrap. New datasets were generated by randomly sampling with replacement from the existing dataset. To each newly sampled dataset all the models were fitted to obtain nonparametric bootstrap samples of ML estimators for each individual transmission parameter from *Appendix 1—table 1*. We then constructed the basic bootstrap 95%-CIs (*Davison and Hinkley, 1997*) as

$$\left(2\widehat{\beta} - q^*_{97.5\%}, 2\widehat{\beta} - q^*_{2.5\%}\right),$$

where $q^*$ denotes the corresponding percentile of the bootstrap sample $\widehat{\beta}^{(1)}, \ldots, \widehat{\beta}^{(B)}$. Finally, we constructed the profile likelihood based CIs (*Held and Bové, 2013*) and compared different types of CIs against the Wald-type CIs (*Appendix 1—figure 11*).

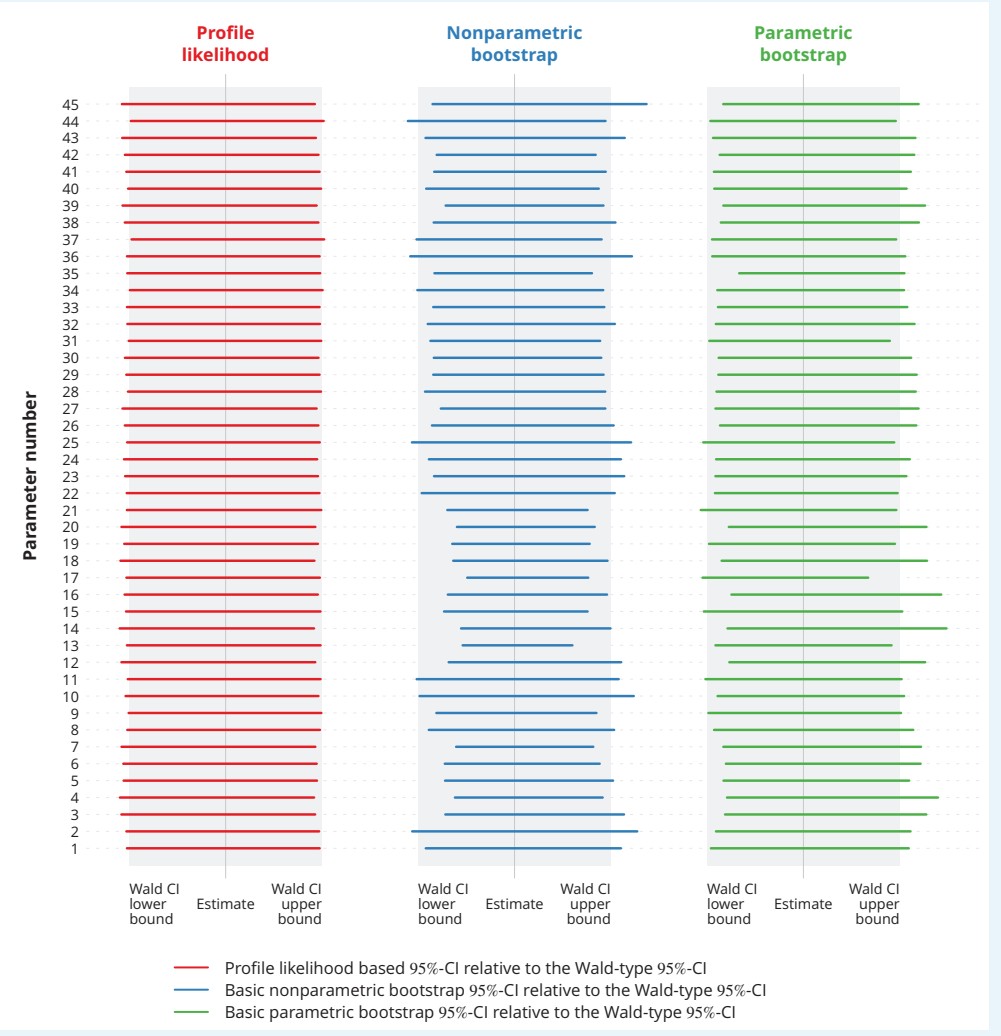

**Appendix 1—figure 11.** Comparison of different types of 95%-confidence intervals (CI) with the normal approximation based Wald-type 95%-CIs. Each column corresponds to a different type of CIs, namely the profile likelihood based CIs, the basic nonparametric bootstrap CIs and the basic parametric bootstrap CIs. Each row represents a single parameter (the overview of the parameters is provided in *Appendix 1—table 1*). The colorful lines show the specific CIs compared to the corresponding Wald-type CIs, namely their relative widths and positions. The gray-shaded areas represent the Wald-type 95%-CIs.
DOI: https://doi.org/10.7554/eLife.28721.023

These simulations indicated no significant difference between the widths of Wald-type and profile likelihood based CIs. Besides, the Wald-type CIs did not appear to be systematically wider or narrower compared to the bootstrap CIs.
To summarize, these simulations imply that the normal approximation Wald-type CIs used in our study provide a reliable alternative to other more time-complex types of CIs.

## Appendix 2

DOI: https://doi.org/10.7554/eLife.28721.024

# Selection of the predictive models

## Single determinant models

To construct a multivariate predictive model for $R_0$ we first focused on each single determinant. More precisely, to find a best predictive model for a single factor we performed both forward selection and backward elimination based on the AIC and BIC criteria (see *Appendix 2—table 1* for the case of establishment date). All terms which appeared in at least one of the single determinant models were later used in the multivariate model.

**Appendix 2—table 1.** Establishment date models obtained with the AIC/BIC forward selection and backward elimination and their respective AIC and BIC values as well as the $p$-values from the likelihood ratio test compared to the null model without any covariates. Terms that were part of the respective final model are marked by ×.

| | AIC | | BIC | |
|---|---|---|---|---|
| | **Forward** | **Backward** | **Forward** | **Backward** |
| $\dfrac{Date_{infection} - 1.1.1996}{365 \cdot 10}$ | | | | |
| $\left(\dfrac{Date_{infection} - 1.1.1996}{365 \cdot 10}\right)^2$ | | × | | × |
| $\left(\dfrac{Date_{infection} - 1.1.1996}{365 \cdot 10}\right)^3$ | × | × | × | × |
| $\left(\dfrac{Date_{infection} - 1.1.1996}{365 \cdot 10}\right)^4$ | × | | × | |
| AIC | 3364.3 | 3364.2 | 3364.3 | 3364.2 |
| BIC | 3382.4 | 3382.3 | 3382.4 | 3382.3 |
| $p$-value from LR test | <0.0001 | <0.0001 | <0.0001 | <0.0001 |

DOI: https://doi.org/10.7554/eLife.28721.025

We chose the model obtained with the backward elimination procedure as the predictive model based solely on the establishment date (*Figure 2*). It provided both the lowest BIC and AIC value, therefore indicating the best goodness-of-fit (*Appendix 2—table 1*).

## Multiple determinants model

Using the terms obtained in the single determinant predictive models (establishment date, age at infection, earliest CD4 cell count, frequency of reporting sex with occasional partner and time to diagnosis) and a viral subtype indicator, we constructed the final multiple determinants model for the prediction as follows. Like before, we carried out both forward and backward selection algorithms for both criteria. Among the resulting algorithms we picked the one minimizing the BIC, since the BIC penalizes the model complexity stronger than the AIC (*Appendix 2—table 2*).

**Appendix 2—table 2.** Multivariate models obtained with the AIC/BIC forward selection and backward elimination algorithms. The terms listed in the table are the terms identified from the single determinant model selections and the crosses indicate the terms entering the multivariate models. The null model from the likelihood ratio test refers to the baseline model without any covariates (not even the subtype).

| | AIC | | BIC | |
|---|---|---|---|---|
| | **Forward** | **Backward** | **Forward** | **Backward** |
| *Subtype* | × | × | × | × |
| $\left(\dfrac{Date_{infection} - 1.1.1996}{365 \cdot 10}\right)^2$ | | | | |
| $\left(\dfrac{Date_{infection} - 1.1.1996}{365 \cdot 10}\right)^3$ | × | × | × | × |
| $\left(\dfrac{Date_{infection} - 1.1.1996}{365 \cdot 10}\right)^4$ | × | × | × | |
| $Rate_{risk}$ | × | × | × | × |
| $\sqrt{Rate_{risk}}$ | × | × | × | |
| $\dfrac{1}{10}\log\left(\dfrac{Years_{diagnosis}}{3}\right)$ | | × | | × |
| $\dfrac{\sqrt{Years_{diagnosis}} - \sqrt{3}}{\sqrt{10}}$ | × | | × | |
| $\dfrac{Years_{diagnosis} - 3}{10}$ | × | × | × | |
| $\left(\dfrac{\sqrt{Years_{diagnosis}} - \sqrt{3}}{\sqrt{10}}\right)^3$ | | × | | × |
| $\dfrac{\sqrt{CD4} - \sqrt{350}}{10}$ | | | | |
| $\left(\dfrac{Age - 32}{10}\right)^2$ | | | | |
| AIC | 3254 | 3252 | 3254 | 3262 |
| BIC | 3314 | 3331 | 3314 | 3316 |
| *p*-value from LR test | <0.0001 | <0.0001 | <0.0001 | <0.0001 |

DOI: https://doi.org/10.7554/eLife.28721.026

## Appendix 3

DOI: https://doi.org/10.7554/eLife.28721.027

# Detailed derivation of the transmission chain size model and statistical inference

## Transmission chain size model

Transmission chains can naturally be modeled as branching processes. The index case corresponds to the root of the process; each new infection represents a new offspring. The generation of an individual in a transmission chain can therefore be interpreted as the transmission degree relative to the index case - the first generation individuals got infected directly from the index case, the second generation indirectly through one mediator, etc. In other words, the transmission degree of a patient is the number of transmission events needed to transfer the virus to this patient from the index case.

## Towards probability generating function of the transmission chain size

Let $R_{k,n}$ denote the number of secondary infections with transmission degree $n$ produced by the $k$th individual from the preceding generation, $S_n$ the total number of new infections of transmission degree $n$ and $Q_N$ the cumulative number of cases in the transmission chain with the transmission degree at most $N$, that is,

$$S_n = \sum_{k=1}^{S_{n-1}} R_{k,n},$$

$$Q_N = \sum_{n=0}^{N} S_n = Q_{N-1} + S_N.$$

The index case establishes the transmission chain and corresponds to the generation $0$, therefore $S_0 = Q_0 = 1$.

Assuming that the numbers of secondary infections are independent and identically distributed for all patients of the same transmission degree, let $\mathcal{R}_n$ denote the probability generating function (PGF) of $R_{k,n}$, namely

$$\mathcal{R}_n(z) := \mathbb{E}\left[z^{R_{k,n}}\right]$$

for each $k \in \{1, 2, \ldots, S_{n-1}\}$. The expected number of secondary infections of degree $n$ is therefore given by

$$\mathbb{E}\left[R_{k,n}\right] = \frac{\mathrm{d}}{\mathrm{d}z}\mathcal{R}_n(z)\bigg|_{z=1} = \mathcal{R}_n^{(1)}(1).$$

Furthermore, assume that the numbers of secondary infections caused by different individuals are independent between each other regardless of the transmission degree. The PGF $\mathcal{Q}_N$ of $Q_N$ is

$$\mathcal{Q}_N(z) := \mathbb{E}\left[z^{Q_N}\right] = \mathbb{E}\left[z^{Q_{N-1}}z^{S_N}\right] \stackrel{(a)}{=} \mathbb{E}\left[\mathbb{E}\left[z^{Q_{N-1}}z^{S_N}|\{S_n\}_{n=0}^{N-1}\right]\right]$$

$$\stackrel{(b)}{=} \mathbb{E}\left[z^{Q_{N-1}}\mathbb{E}\left[z^{S_N}|\{S_n\}_{n=0}^{N-1}\right]\right] = \mathbb{E}\left[z^{Q_{N-1}}\mathbb{E}\left[\prod_{k=1}^{S_{N-1}}z^{R_{k,N}}|\{S_n\}_{n=0}^{N-1}\right]\right]$$

$$\stackrel{(c)}{=} \mathbb{E}\left[z^{Q_{N-1}}\prod_{k=1}^{S_{N-1}}\mathbb{E}\left[z^{R_{k,N}}|\{S_n\}_{n=0}^{N-1}\right]\right]$$

$$\stackrel{(d)}{=} \mathbb{E}\left[z^{Q_{N-1}}\prod_{k=1}^{S_{N-1}}\mathbb{E}\left[z^{R_{k,N}}\right]\right] = \mathbb{E}\left[z^{Q_{N-1}}\prod_{k=1}^{S_{N-1}}\mathcal{R}_N(z)\right]$$

$$= \mathbb{E}\left[z^{Q_{N-1}}\mathcal{R}_N(z)^{S_{N-1}}\right],$$

because (a) of the tower property of the conditional expectation, (b) $Q_{N-1} = \sum_{k=0}^{N-1}S_n$ is $\{S_n\}_{n=0}^{N-1}$-measurable, (c) $\{R_{k,N}\}_{k=1}^{S_{N-1}}$ are independent, and (d) $R_{k,N}$ are independent from $\{S_n\}_{n=0}^{N-1}$ for all $k = 1, 2, \ldots, S_{N-1}$. Repeating similar steps iteratively yields

$$\mathcal{Q}_N(z) = \mathbb{E}\left[z^{Q_{N-2}}z^{S_{N-1}}\mathcal{R}_N(z)^{S_{N-1}}\right] = \mathbb{E}\left[z^{Q_{N-2}}\mathbb{E}\left[(z\mathcal{R}_N(z))^{S_{N-1}}|\{S_n\}_{n=0}^{N-2}\right]\right] \quad (1)$$

$$= \mathbb{E}\left[z^{Q_{N-3}}z^{S_{N-2}}\mathcal{R}_{N-1}(z\mathcal{R}_N(z))^{S_{N-2}}\right] = \ldots$$

$$= \mathbb{E}\left[z^{Q_{N-4}}(z\mathcal{R}_{N-2}(z\mathcal{R}_{N-1}(z\mathcal{R}_N(z))))^{S_{N-3}}\right] = \ldots$$

$$\vdots$$

$$= \mathbb{E}\left[(z\mathcal{R}_1(z\mathcal{R}_2(\ldots(z\mathcal{R}_N(z)\ldots))))^{S_0}\right]$$

$$= z\mathcal{R}_1(z\mathcal{R}_2(\ldots(z\mathcal{R}_N(z)\ldots))).$$

The total size of the transmission chain is denoted by $T$ and equals

$$T := \lim_{N\to\infty} Q_N.$$

From the definition of $T$ it follows that its PGF $\mathcal{T}$ equals

$$\mathcal{T}(z) = \lim_{N\to\infty} \mathcal{Q}_N(z) \quad (2)$$

for all $z$.

**Probability generating function of a completely observed uniform transmission chain**

Assume that the number of secondary infections follows the same distribution with PGF $\mathcal{G}$ for all infected persons, namely

$$\mathcal{R}_n \equiv \mathcal{G}$$

for every $n$ (i.e., the transmission is uniform across different transmission degrees). The PGF $\mathcal{Q}_N$ (**Equation 1**) then simplifies to

$$\mathcal{Q}_1(z) = z\mathcal{G}(z)$$
$$\mathcal{Q}_2(z) = z\mathcal{G}(z\mathcal{G}(z)) = z\mathcal{G}(\mathcal{Q}_1(z))$$
$$\vdots$$
$$\mathcal{Q}_N(z) = z\mathcal{G}(\mathcal{Q}_{N-1}(z)).$$

Using **Equation 2**, the PGF $\mathcal{T}$ for each $z$ solves the equation

$$\mathcal{T}(z) \stackrel{!}{=} z\mathcal{G}(\mathcal{T}(z)).$$

**Probability generating function of a transmission chain with modified transmission potential of the index case**

From the perspective of the Swiss HIV heterosexual population the index case might have lost some of its potential to transmit the virus prior to establishing the transmission chain in the

population under consideration. The follow-up cases are infected while already in the subpopulation and can therefore fully contribute to spreading. Sex workers and lonely foreigners in Switzerland represent two examples of index cases with an enhanced transmission potential. We assume that apart from the index case the numbers of secondary infections are equally and independently distributed for all the other infected individuals. Let $\rho_{\text{index}}$ denote the *index case relative transmission potential (ICRTP)*. In terms of the model the above assumptions can be summarized as

$$\mathcal{R}_1(z) = \mathcal{F}(z),$$
$$\mathcal{R}_n(z) = \mathcal{G}(z), \qquad \text{for } n>1,$$

where $\mathcal{F}$ and $\mathcal{G}$ denote the PGF of two distributions, such that

$$\mathcal{F}^{(1)}(1) = \rho_{\text{index}}\mathcal{G}^{(1)}(1),$$

namely $\mathbb{E}[R_{1,1}] = \rho_{\text{index}}\mathbb{E}[R_{k,n}]$ for all $k \in \{1, \ldots, S_{n-1}\}$ and $n>1$. In other words, the ICRTP is the expected number of secondary infections of the index case relative to the expected number of secondary infections of the rest of the transmission chain.

To compute the PGF of the transmission chain with modified transmissibility of the index case we first introduce a *skeleton* function $\mathcal{K}$, which controls the regular part/tail of the transmission chain. Let $\mathcal{K}$ be the pointwise limit $\mathcal{K}(z) := \lim_{N\to\infty} \mathcal{K}_N(z)$ of the iteratively defined functions

$$\mathcal{K}_1(z) := z$$
$$\mathcal{K}_N(z) := z\mathcal{G}(\mathcal{K}_{N-1}(z)).$$

The skeleton therefore solves the equation

$$\mathcal{K}(z) \stackrel{!}{=} z\mathcal{G}(\mathcal{K}(z)). \qquad (3)$$

Note that in the absence of the modified transmissibility of the index case, the skeleton function $\mathcal{K}$ coincides with the PGF of the transmission chain size. Having introduced this notation one can rewrite the PGF $\mathcal{Q}_N$ (**Equation 1**) as

$$\mathcal{Q}_1(z) = z\mathcal{F}(z) = z\mathcal{F}(\mathcal{K}_1(z))$$
$$\mathcal{Q}_2(z) = z\mathcal{F}(z\mathcal{G}(z)) = z\mathcal{F}(\mathcal{K}_2(z))$$
$$\mathcal{Q}_3(z) = z\mathcal{F}(z\mathcal{G}(z\mathcal{G}(z))) = z\mathcal{F}(\mathcal{K}_3(z))$$
$$\vdots$$
$$\mathcal{Q}_N(z) = z\mathcal{F}(\mathcal{K}_N(z)).$$

As $N \to \infty$, this implies

$$\mathcal{T}(z) = z\mathcal{F}(\mathcal{K}(z)) \qquad (4)$$

for all $z$.

### Probability generating function of an incompletely observed transmission chain

Since not every HIV infected person is included in a cohort, linked to care or even diagnosed, we only observe parts of the transmission chains. Suppose that each infection is detected with probability $p$, independently of the others. Furthermore, assume that despite not all cases being observed, the sampled patients belonging to the same true transmission chain could be identified as members of this transmission cluster (and not as members of two or more separate transmission clusters).

The true transmission chain can still be modeled with the branching process as above. Let tilde ($\tilde{\ }$) denote the observed cases. Since each case is detected at random with probability $p$ the following applies to the observed transmission chains.

- If $R_{k,n}$ is defined as above then $\widetilde{R}_{k,n}$ denotes the number of secondary infections with transmission degree $n$ caused by patient $k$ which are actually observed. It follows

$$\widetilde{R}_{k,n}|R_{k,n} \sim Bin(R_{k,n},p).$$

- Given the numbers of secondary infections with transmission degree $n$ of all the patients the observed number of infections of transmission degree $n$ equals

$$\widetilde{S}_n = \sum_{k=1}^{S_{n-1}} \widetilde{R}_{k,n}$$

and follows a binomial distribution, namely

$$\widetilde{S}_n|\{R_{k,n}\}_{k=1}^{S_{n-1}} \sim \sum_{k=1}^{S_{n-1}} Bin(R_{k,n},p) = Bin\left(\sum_{k=1}^{S_{n-1}} R_{k,n},p\right) = Bin(S_n,p).$$

- The observed cumulative number of infected individuals with the transmission degree at most $N$ equals

$$\widetilde{Q}_N = \sum_{n=0}^{N} \widetilde{S}_n = \widetilde{Q}_{N-1} + \widetilde{S}_N = \widetilde{Q}_{N-1} + \sum_{k=1}^{S_{N-1}} \widetilde{R}_{k,N}.$$

By conditioning on the cumulative number of infections of transmission degree up to $N-1$ and on the numbers of secondary infections of transmission degree $N$, $\widetilde{Q}_N$ therefore follows a binomial distribution, that is,

$$\widetilde{Q}_N|Q_{N-1},\{R_{k,N}\}_{k=1}^{S_{N-1}} \sim Bin\left(Q_{N-1} + \sum_{k=1}^{S_{N-1}} R_{k,N},p\right) = Bin(Q_{N-1}+S_N,p) = Bin(Q_N,p).$$

Since $\mathcal{B}(z) = ((1-p)+pz)^n$ is the PGF of a $Bin(n,p)$-distributed random variable, the PGF of $\widetilde{Q}_N$ can be expressed as

$$
\begin{aligned}
\widetilde{Q}_N(z) = \mathbb{E}\left[z^{\widetilde{Q}_N}\right] &\overset{(a)}{=} \mathbb{E}\left[\mathbb{E}\left[z^{\widetilde{Q}_N}|Q_N\right]\right] \\
&\overset{(b)}{=} \mathbb{E}\left[\mathbb{E}\left[\mathbb{E}\left[z^{\widetilde{Q}_N}|Q_{N-1},\{R_{k,N}\}_{k=1}^{S_{N-1}}\right]|Q_N\right]\right] \\
&\overset{(c)}{=} \mathbb{E}\left[\mathbb{E}\left[((1-p)+pz)^{Q_{N-1}+\sum_{k=1}^{S_{N-1}}R_{k,N}}|Q_N\right]\right] = \mathbb{E}\left[\mathbb{E}\left[((1-p)+pz)^{Q_N}|Q_N\right]\right] \\
&\overset{(a)}{=} \mathbb{E}\left[((1-p)+pz)^{Q_N}\right] \\
&= \mathcal{Q}_N((1-p)+pz)
\end{aligned}
$$

in terms of the PGF $\mathcal{Q}_N$, because (a) of the tower property, (b) of the tower property for $\sigma$-algebras $\sigma(Q_N) \subseteq \sigma\left(Q_{N-1},\{R_{k,N}\}_{k=1}^{S_{N-1}}\right)$ due to the relation $Q_N = Q_{N-1} + \sum_{k=1}^{S_{N-1}} R_{k,N}$, and (c) $Q_N$ given $Q_{N-1}$ and $\{R_{k,N}\}_{k=1}^{S_{N-1}}$ is binomially distributed.

This allows us to obtain the PGF $\widetilde{\mathcal{T}}$ of the observed transmission chain length $\widetilde{T} = \lim_{N\to\infty} \widetilde{Q}_N$, namely

$$\widetilde{\mathcal{T}}(z) = \mathcal{T}((1-p)+pz),$$

where $\mathcal{T}$ denotes the PGF of the true underlying transmission chain.
Finally, the PGF of the observed transmission chain size with modified transmissibility of the index case equals

$$\widetilde{\mathcal{T}}(z) = \mathcal{T}((1-p)+pz) = ((1-p)+pz)\mathcal{F}(\mathcal{K}((1-p)+pz)). \qquad (5)$$

## Inferring the transmission parameters

Probability generating functions enable us to obtain the state probabilities, namely the probability of observing a transmission chain of length $j$ can be calculated as

$$\mathbb{P}\big[\widetilde{T}=j\big] = \frac{\widetilde{\mathcal{T}}^{(j)}(0)}{j!},$$

where $^{(j)}$ denotes the $j$th derivative. The transmission chains with no observed cases are not observable, therefore we are interested in the probability that an observed chain is of length $j$, which equals

$$\mathbb{P}\big[\widetilde{T}=j\,\big|\,\widetilde{T}>0\big] = \frac{\mathbb{P}\big[\widetilde{T}=j\big]}{1 - \mathbb{P}\big[\widetilde{T}=0\big]} = \frac{1}{j!}\cdot\frac{\widetilde{\mathcal{T}}^{(j)}(0)}{1 - \widetilde{\mathcal{T}}(0)}.$$

So far, we have not included the basic reproductive number or any other transmission-related parameters in the PGF of transmission chain size $\widetilde{\mathcal{T}}$. In the following paragraphs we extensively present the statistical inference (following **Held and Bové, 2013**) of the transmission parameters based on the transmission chain size model described above.

**The likelihood function**

Let $\boldsymbol{\omega}$ denote a vector of transmission parameters, for instance $\boldsymbol{\omega} = R_0$ in case of a single transmission parameter corresponding to the basic reproductive number. Assuming that the transmission chain sizes are independent, the likelihood function of the sample of $I$ observed transmission chain sizes $\widetilde{\mathbf{t}} := \{\widetilde{t}_i\}_{i=1}^{I}$ is defined by

$$L_{\boldsymbol{\omega}}\big(\boldsymbol{\omega}|\widetilde{\mathbf{t}}\big) := \prod_{i=1}^{I} \frac{1}{\widetilde{t}_i!}\cdot\frac{\widetilde{\mathcal{T}}^{(\widetilde{t}_i)}(0;\boldsymbol{\omega})}{1 - \widetilde{\mathcal{T}}(0;\boldsymbol{\omega})},$$

where $\widetilde{\mathcal{T}}(z;\boldsymbol{\omega})$ denotes the PGF of transmission chain size with transmission parameters $\boldsymbol{\omega}$. The corresponding log-likelihood function is

$$\ell_{\boldsymbol{\omega}}\big(\boldsymbol{\omega}|\widetilde{\mathbf{t}}\big) = \sum_{i=1}^{I}\left(\log\left(\widetilde{\mathcal{T}}^{(\widetilde{t}_i)}(0;\boldsymbol{\omega})\right) - \log(\widetilde{t}_i!) - \log\left(1 - \widetilde{\mathcal{T}}(0;\boldsymbol{\omega})\right)\right). \tag{6}$$

Since the transmission parameters are often required to be positive the log-parameterization is more appropriate. Let $\boldsymbol{\theta}$ denote the transmission parameters on the logarithmic scale, namely $\boldsymbol{\theta} := \log(\boldsymbol{\omega})$. The log-parameterized log-likelihood function is therefore $\ell(\boldsymbol{\theta}|\widetilde{\mathbf{t}}) := \ell_{\boldsymbol{\omega}}(\log(\boldsymbol{\omega})|\widetilde{\mathbf{t}})$. The Jacobian matrix corresponding to the log-parameterization equals

$$\mathbf{J}_{\boldsymbol{\omega}(\boldsymbol{\theta})} = diag\big(e^{\boldsymbol{\theta}}\big) = diag(\boldsymbol{\omega}),$$

where $diag(x)$ denotes a diagonal matrix with vector $x$ representing its diagonal elements.

**The score function and the Fisher information matrix**

The maximum likelihood (ML) estimator $\widehat{\boldsymbol{\omega}}$ maximizes the log-likelihood function and is a root of the score function

$$\mathbf{u}_{\boldsymbol{\omega}}\big(\boldsymbol{\omega}|\widetilde{\mathbf{t}}\big) := \frac{\partial}{\partial\boldsymbol{\omega}}\ell_{\boldsymbol{\omega}}\big(\boldsymbol{\omega}|\widetilde{\mathbf{t}}\big) = \sum_{i=1}^{I}\left(\frac{\frac{\partial}{\partial\boldsymbol{\omega}}\widetilde{\mathcal{T}}^{(\widetilde{t}_i)}(0;\boldsymbol{\omega})}{\widetilde{\mathcal{T}}^{(\widetilde{t}_i)}(0;\boldsymbol{\omega})} + \frac{\frac{\partial}{\partial\boldsymbol{\omega}}\widetilde{\mathcal{T}}(0;\boldsymbol{\omega})}{1 - \widetilde{\mathcal{T}}(0;\boldsymbol{\omega})}\right),$$

or equivalently, the ML estimator $\widehat{\boldsymbol{\theta}}$ solves

$$\mathbf{u}\big(\widehat{\boldsymbol{\theta}}|\widetilde{\mathbf{t}}\big) = \mathbf{J}_{\boldsymbol{\omega}(\boldsymbol{\theta})}^{T}\mathbf{u}_{\boldsymbol{\omega}}\big(e^{\widehat{\boldsymbol{\theta}}}|\widetilde{\mathbf{t}}\big) \stackrel{!}{=} \mathbf{0},$$

where $\mathbf{u}$ denotes the score function corresponding to the log-parameterized log-likelihood $\ell$. The Fisher information matrix $\mathcal{I}_{\boldsymbol{\omega}}\big(\boldsymbol{\omega}|\widetilde{\mathbf{t}}\big) := -\frac{\partial^2}{\partial\boldsymbol{\omega}^2}\ell_{\boldsymbol{\omega}}\big(\boldsymbol{\omega}|\widetilde{\mathbf{t}}\big)$ is given by

$$\mathcal{I}_{\boldsymbol{\omega}}(\boldsymbol{\omega}|\widetilde{\mathbf{t}}) = -\sum_{i=1}^{I}\left(\frac{\frac{\partial^2}{\partial\boldsymbol{\omega}^2}\widetilde{\mathcal{T}}^{(\widetilde{t}_i)}(0;\boldsymbol{\omega})}{\widetilde{\mathcal{T}}^{(\widetilde{t}_i)}(0;\boldsymbol{\omega})} - \left(\frac{\frac{\partial}{\partial\boldsymbol{\omega}}\widetilde{\mathcal{T}}^{(\widetilde{t}_i)}(0;\boldsymbol{\omega})}{\widetilde{\mathcal{T}}^{(\widetilde{t}_i)}(0;\boldsymbol{\omega})}\right)^2 + \frac{\frac{\partial^2}{\partial\boldsymbol{\omega}^2}\widetilde{\mathcal{T}}(0;\boldsymbol{\omega})}{1-\widetilde{\mathcal{T}}(0;\boldsymbol{\omega})} + \left(\frac{\frac{\partial}{\partial\boldsymbol{\omega}}\widetilde{\mathcal{T}}(0;\boldsymbol{\omega})}{1-\widetilde{\mathcal{T}}(0;\boldsymbol{\omega})}\right)^2\right)$$

and equals

$$\mathcal{I}(\boldsymbol{\theta}|\widetilde{\mathbf{t}}) = \mathbf{J}_{\boldsymbol{\omega}(\boldsymbol{\theta})}^{T}\mathcal{I}_{\boldsymbol{\omega}}\left(e^{\boldsymbol{\theta}}|\widetilde{\mathbf{t}}\right)\mathbf{J}_{\boldsymbol{\omega}(\boldsymbol{\theta})} - diag\left(\mathbf{u}_{\boldsymbol{\omega}}\left(e^{\boldsymbol{\theta}}|\widetilde{\mathbf{t}}\right)\right)\mathbf{J}_{\boldsymbol{\omega}(\boldsymbol{\theta})}$$

under the log-parameterization due to the chain rule in higher dimensions and the special form of the transformation corresponding to the log-parameterization. The PGF function $\widetilde{\mathcal{T}}$ and its derivatives are thus crucial (and sufficient) for the statistical inference, since the log-likelihood function $\ell$, the score function $\mathbf{u}$ and the Fisher information matrix $\mathcal{I}$ can be expressed in terms of $\widetilde{\mathcal{T}}$ only.

**Confidence intervals and hypothesis testing**
Assuming that the regularity conditions are satisfied (**Held and Bové, 2013**) the ML estimator is unbiased and asymptotically normally distributed with variance equal to the inverse observed Fisher information matrix. Hence, for each parameter $\theta \in \boldsymbol{\theta}$ we can construct the Wald $\alpha\%$-confidence interval as

$$\mathcal{C}_{\theta,\alpha} = \left(\widehat{\theta} - z_{\frac{1+\alpha}{2}}se\left(\widehat{\theta}\right), \widehat{\theta} + z_{\frac{1+\alpha}{2}}se\left(\widehat{\theta}\right)\right)$$

where $z_{\frac{1+\alpha}{2}}$ denotes the $\frac{1+\alpha}{2}$-quantile of the standard normal distribution, and the standard error $se\left(\widehat{\theta}\right)$ is defined as

$$se\left(\widehat{\theta}\right) := \sqrt{\mathcal{I}^{-1}\left(\widehat{\boldsymbol{\theta}}|\widetilde{\mathbf{t}}\right)_{\theta\theta}},$$

and $_{\theta\theta}$ denotes the diagonal element of the inversed observed Fisher information matrix $\mathcal{I}(\boldsymbol{\theta}|\widetilde{\mathbf{t}})$ corresponding to parameter $\theta$. The approximate $\alpha\%$-confidence interval for the original parameter $\omega$ is obtained by the reverse transformation

$$\mathcal{C}_{\omega,\alpha} = e^{\mathcal{C}_{\theta,\alpha}}.$$

Similarly, to test the hypothesis $H_0 : \theta = \theta_0$ against the alternative $H_A$, the Wald test statistic

$$\tau_\theta(\theta_0) := \frac{\widehat{\theta} - \theta_0}{se\left(\widehat{\theta}\right)}$$

can be used. Assuming the standard normal distribution of the test statistic under null hypothesis, the *p*-value equals

- $2 \cdot (1 - \Phi(|\tau_\theta(\theta_0)|))$ for the alternative hypothesis $H_A : \theta \neq \theta_0$,
- $\Phi(\tau_\theta(\theta_0))$ for the alternative $H_A : \theta < \theta_0$, and
- $1 - \Phi(\tau_\theta(\theta_0))$ for the alternative $H_A : \theta > \theta_0$;

where $\Phi$ is the cumulative distribution function of the standard normal distribution.

## Generalized transmission chain size model
Suppose that the variability of one of the parameters can be explained through a linear combination of different covariates, namely

$$\boldsymbol{\theta}_i := \left(\boldsymbol{\beta}^T\mathbf{x}_i, \boldsymbol{\eta}\right)$$

are the transmission parameters of the $i$th chain with characteristics $\mathbf{x}_i$, where $\boldsymbol{\eta}$ denotes the remaining parameters from $\boldsymbol{\theta}$ which are not modeled as a linear combination. Furthermore, it is plausible to assume that while the transmission chains share all the transmission parameters $(\boldsymbol{\beta}, \boldsymbol{\eta})$, their transmission chain size distribution may differ due to different sampling densities

or different offspring distribution of the index case (for instance, for the transmission chains originating from other Swiss transmission groups the ICRTP is irrelevant/equals $\rho_{\text{index}} = 1$). Let $\widetilde{\mathcal{T}}_i$ be the PGF corresponding to the transmission chain $i$ and let $\mathbf{X} := \{\mathbf{x}_i\}_{i=1}^I$. The generalized log-likelihood function is hence given by

$$\ell(\boldsymbol{\beta}, \boldsymbol{\eta} | \widetilde{\mathbf{t}}, \mathbf{X}) = \sum_{i=1}^{I} \left( \log\left( \widetilde{\mathcal{T}}_i^{(\widetilde{t}_i)}(0; \boldsymbol{\omega}_i(\boldsymbol{\beta}, \boldsymbol{\eta})) \right) - \log(\widetilde{t}_i!) - \log\left( 1 - \widetilde{\mathcal{T}}_i(0; \boldsymbol{\omega}_i(\boldsymbol{\beta}, \boldsymbol{\eta})) \right) \right),$$

where

$$\boldsymbol{\omega}_i(\boldsymbol{\beta}, \boldsymbol{\eta}) := \left( e^{\boldsymbol{\beta}^T \mathbf{x}_i}, e^{\boldsymbol{\eta}} \right).$$

The corresponding Jacobian matrix equals

$$\mathbf{J}_{\boldsymbol{\omega}_i}(\boldsymbol{\beta}, \boldsymbol{\eta}) = \begin{bmatrix} e^{\boldsymbol{\beta}^T \mathbf{x}_i} \mathbf{x}_i^T & \mathbf{0} \\ \mathbf{0} & diag(e^{\boldsymbol{\eta}}) \end{bmatrix}.$$

In the generalized model, the score function is

$$\mathbf{u}(\boldsymbol{\beta}, \boldsymbol{\eta} | \widetilde{\mathbf{t}}, \mathbf{X}) := \frac{\partial}{\partial(\boldsymbol{\beta}, \boldsymbol{\eta})} \ell(\boldsymbol{\beta}, \boldsymbol{\eta} | \widetilde{\mathbf{t}}, \mathbf{X}) = \sum_{i=1}^{I} \mathbf{J}_{\boldsymbol{\omega}_i}(\boldsymbol{\beta}, \boldsymbol{\eta})^T \left( \frac{\frac{\partial}{\partial \boldsymbol{\omega}} \widetilde{\mathcal{T}}_i^{(\widetilde{t}_i)}(0; \boldsymbol{\omega}_i(\boldsymbol{\beta}, \boldsymbol{\eta}))}{\widetilde{\mathcal{T}}_i^{(\widetilde{t}_i)}(0; \boldsymbol{\omega}_i(\boldsymbol{\beta}, \boldsymbol{\eta}))} + \frac{\frac{\partial}{\partial \boldsymbol{\omega}} \widetilde{\mathcal{T}}_i(0; \boldsymbol{\omega}_i(\boldsymbol{\beta}, \boldsymbol{\eta}))}{1 - \widetilde{\mathcal{T}}_i(0; \boldsymbol{\omega}_i(\boldsymbol{\beta}, \boldsymbol{\eta}))} \right)$$

and the Fisher information matrix as

$$\mathcal{I}(\boldsymbol{\beta}, \boldsymbol{\eta} | \widetilde{\mathbf{t}}, \mathbf{X}) := -\frac{\partial^2}{\partial^2(\boldsymbol{\beta}, \boldsymbol{\eta})} \ell(\boldsymbol{\beta}, \boldsymbol{\eta} | \widetilde{\mathbf{t}}, \mathbf{X})$$

$$= -\sum_{i=1}^{I} \mathbf{J}_{\boldsymbol{\omega}_i}(\boldsymbol{\beta}, \boldsymbol{\eta})^T \left( diag\left( \frac{\frac{\partial}{\partial \boldsymbol{\omega}} \widetilde{\mathcal{T}}_i^{(\widetilde{t}_i)}(0; \boldsymbol{\omega}_i(\boldsymbol{\beta}, \boldsymbol{\eta}))}{\widetilde{\mathcal{T}}_i^{(\widetilde{t}_i)}(0; \boldsymbol{\omega}_i(\boldsymbol{\beta}, \boldsymbol{\eta}))} + \frac{\frac{\partial}{\partial \boldsymbol{\omega}} \widetilde{\mathcal{T}}_i(0; \boldsymbol{\omega}_i(\boldsymbol{\beta}, \boldsymbol{\eta}))}{1 - \widetilde{\mathcal{T}}_i(0; \boldsymbol{\omega}_i(\boldsymbol{\beta}, \boldsymbol{\eta}))} \right) \mathbf{J}_{\boldsymbol{\theta}_i}(\boldsymbol{\beta}, \boldsymbol{\eta}) \right.$$

$$+ \left( \frac{\frac{\partial^2}{\partial \boldsymbol{\omega}^2} \widetilde{\mathcal{T}}_i^{(\widetilde{t}_i)}(0; \boldsymbol{\omega}_i(\boldsymbol{\beta}, \boldsymbol{\eta}))}{\widetilde{\mathcal{T}}_i^{(\widetilde{t}_i)}(0; \boldsymbol{\omega})} - \left( \frac{\frac{\partial}{\partial \boldsymbol{\omega}} \widetilde{\mathcal{T}}_i^{(\widetilde{t}_i)}(0; \boldsymbol{\omega}_i(\boldsymbol{\beta}, \boldsymbol{\eta}))}{\widetilde{\mathcal{T}}_i^{(\widetilde{t}_i)}(0; \boldsymbol{\omega}_i(\boldsymbol{\beta}, \boldsymbol{\eta}))} \right)^2 \right.$$

$$\left. \left. + \frac{\frac{\partial^2}{\partial \boldsymbol{\omega}^2} \widetilde{\mathcal{T}}_i(0; \boldsymbol{\omega}_i(\boldsymbol{\beta}, \boldsymbol{\eta}))}{1 - \widetilde{\mathcal{T}}_i(0; \boldsymbol{\omega}_i(\boldsymbol{\beta}, \boldsymbol{\eta}))} + \left( \frac{\frac{\partial}{\partial \boldsymbol{\omega}} \widetilde{\mathcal{T}}_i(0; \boldsymbol{\omega}_i(\boldsymbol{\beta}, \boldsymbol{\eta}))}{1 - \widetilde{\mathcal{T}}_i(0; \boldsymbol{\omega}_i(\boldsymbol{\beta}, \boldsymbol{\eta}))} \right)^2 \right) \mathbf{J}_{\boldsymbol{\omega}_i}(\boldsymbol{\beta}, \boldsymbol{\eta}) \right)$$

### Prediction intervals

It is tempting to construct an approximate confidence interval for the parameter $\theta_i := \boldsymbol{\beta}^T \mathbf{x}_i$. Since the parameter $\theta_i$ is a prediction rather than an estimate, the element of interest is the prediction interval, which takes into account both the characteristics $\mathbf{x}_i$ and the uncertainty of all parameter estimates $\widehat{\boldsymbol{\beta}}$.

Assuming that the ML estimator $\left( \widehat{\boldsymbol{\beta}}, \widehat{\boldsymbol{\eta}} \right)$ is asymptotically normally distributed, it follows that the linear combination $\widehat{\boldsymbol{\beta}}^T \mathbf{x}_i$ is also asymptotically Gaussian, specifically

$$\widehat{\boldsymbol{\beta}}^T \mathbf{x}_i \overset{\text{a.}}{\sim} \mathcal{N}\left( \boldsymbol{\beta}^T \mathbf{x}_i, \mathbf{x}_i^T Var\left( \widehat{\boldsymbol{\beta}} \right) \mathbf{x}_i \right).$$

The variance $Var\left( \widehat{\boldsymbol{\beta}} \right)$ can be approximated by the inverse of the observed Fisher information matrix as

$$Var\left( \widehat{\boldsymbol{\beta}} \right) \approx \mathcal{I}^{-1}\left( \widehat{\boldsymbol{\beta}}, \widehat{\boldsymbol{\eta}} | \widetilde{\mathbf{t}}, \mathbf{X} \right)_{\boldsymbol{\beta}\boldsymbol{\beta}}.$$

Finally, an approximate $\alpha\%$-prediction interval for $\theta_i$ is constructed as

$$\mathcal{P}_{\theta_i,\alpha} = \left(\widehat{\boldsymbol{\beta}}^T\mathbf{x}_i - z_{\frac{1+\alpha}{2}}se\left(\widehat{\boldsymbol{\beta}}^T\mathbf{x}_i\right), \widehat{\boldsymbol{\beta}}^T\mathbf{x}_i + z_{\frac{1+\alpha}{2}}se\left(\widehat{\boldsymbol{\beta}}^T\mathbf{x}_i\right)\right),$$

with

$$se\left(\widehat{\boldsymbol{\beta}}^T\mathbf{x}_i\right) := \sqrt{\mathbf{x}_i^T Var\left(\widehat{\boldsymbol{\beta}}\right)\mathbf{x}_i}.$$

## Example: Poisson model

Suppose that the number of secondary infections follows the Poisson distribution with parameter $R_0$. Taking into account the modified transmissibility of the index case $\rho_{\text{index}}$ (wherever applicable), the PGFs $\mathcal{F}$ and $\mathcal{G}$ for the index case and the tail, respectively, are

$$\mathcal{F}(z;R_0) = e^{\rho_{\text{index}}R_0(z-1)},$$
$$\mathcal{G}(z;R_0) = e^{R_0(z-1)}.$$

The skeleton function $\mathcal{K}$ thus solves

$$\mathcal{K}(z;R_0) \stackrel{!}{=} ze^{R_0(\mathcal{K}(z;R_0)-1)}, \qquad \forall z. \tag{7}$$

Consider an imperfectly sampled transmission chain with probability of detection $p$ and with ICRTP $\rho_{\text{index}}$. The aim is to obtain the Taylor coefficients of $\widetilde{\mathcal{T}}$ around $z=0$ to be able to estimate the transmission parameter $R_0$ with the maximum likelihood approach since they are needed to calculate the log-likelihood (**Equation 6**).
Let

$$w := (1-p) + pz$$

and

$$\mathcal{Y}(w;R_0) := \widetilde{\mathcal{T}}\left(\frac{w - (1-p)}{p};R_0\right)$$

such that $\mathcal{Y}((1-p) + pz;R_0) = \widetilde{\mathcal{T}}(z;R_0)$ and that the **Equation 5** of the PGF of observed transmission chain size $\widetilde{\mathcal{T}}$ simplifies to

$$\mathcal{Y}(w;R_0) = w\mathcal{F}(\mathcal{K}(w;R_0)).$$

Taking into account the PGF $\mathcal{F}$ of the index case implies

$$\mathcal{Y}(w;R_0) = we^{\rho_{\text{index}}R_0(\mathcal{K}(w;R_0)-1)}.$$

Solving for $\mathcal{K}(w;R_0)$ yields

$$\mathcal{K}(w;R_0) = \frac{1}{\rho_{\text{index}}R_0}\log\left(\frac{\mathcal{Y}(w;R_0)}{w}\right) + 1.$$

Plugging this into **Equation 7** gives

$$\frac{1}{\rho_{\text{index}}R_0}\log\left(\frac{\mathcal{Y}(w;R_0)}{w}\right) + 1 = we^{\frac{1}{\rho_{\text{index}}}\log\left(\frac{\mathcal{Y}(w;R_0)}{w}\right)}$$
$$\frac{1}{\rho_{\text{index}}R_0}\log\left(\frac{\mathcal{Y}(w;R_0)}{w}\right) = w\left(\frac{\mathcal{Y}(w;R_0)}{w}\right)^{\frac{1}{\rho_{\text{index}}}} - 1.$$

With $\mathcal{Z}(w;R_0) := \left(\frac{\mathcal{Y}(w;R_0)}{w}\right)^{\frac{1}{\rho_{\text{index}}}}$, the last equation is equivalent to

$$\frac{1}{R_0}\log(\mathcal{Z}(w;R_0)) = w\mathcal{Z}(w;R_0) - 1$$

$$\mathcal{Z}(w;R_0) = e^{R_0(w\mathcal{Z}(w;R_0)-1)}$$

$$\mathcal{Z}(w;R_0)e^{-R_0 w\mathcal{Z}(w;R_0)} = e^{-R_0}$$

$$-R_0 w\mathcal{Z}(w;R_0)e^{-R_0 w\mathcal{Z}(w;R_0)} = -R_0 w e^{-R_0},$$

which is an equation of the form $f(w)e^{f(w)} = g(w)$. The latter admits a solution $f(w) = W_0(g(w))$, where $W_0$ is the principal branch of the Lambert $W$ function (**Corless et al., 1996**). Thus

$$\mathcal{Z}(w;R_0) = \frac{W_0(-R_0 w e^{-R_0})}{-R_0 w},$$

and finally,

$$\mathcal{Y}(w;R_0) = w\mathcal{Z}(w;R_0)^{\rho_{\text{index}}} = w\left(\frac{W_0(-R_0 w e^{-R_0})}{-R_0 w}\right)^{\rho_{\text{index}}} = w\left(e^{-R_0}\frac{W_0(-R_0 w e^{-R_0})}{-R_0 w e^{-R_0}}\right)^{\rho_{\text{index}}}$$

$$= w e^{-\rho_{\text{index}}R_0}\left(\frac{W_0(-R_0 w e^{-R_0})}{-R_0 w e^{-R_0}}\right)^{\rho_{\text{index}}}.$$

Using the relation $\frac{W_0(-x)}{-x} = e^{-W_0(-x)}$ (which follows from the definition of $W_0(-x)$), we have

$$\mathcal{Y}(w;R_0) = w e^{-\rho_{\text{index}}R_0}e^{-\rho_{\text{index}}W_0(-R_0 w e^{-R_0})}.$$

From the Taylor expansion of $e^{-\gamma W_0(-x)} = \sum_{m=0}^{\infty}\gamma(\gamma+m)^{m-1}\frac{x^m}{m!}$ around $x = 0$ (equality (2.36) in **Corless et al., 1996**), we obtain

$$\mathcal{Y}(w;R_0) = w e^{-\rho_{\text{index}}R_0}\sum_{m=0}^{\infty}\frac{\rho_{\text{index}}(m+\rho_{\text{index}})^{m-1}}{m!}\left(R_0 w e^{-R_0}\right)^m$$

$$= \sum_{m=0}^{\infty}\frac{\rho_{\text{index}}(m+\rho_{\text{index}})^{m-1}R_0^m e^{-R_0(m+\rho_{\text{index}})}}{m!}w^{m+1}$$

$$= \sum_{m=1}^{\infty}\frac{\rho_{\text{index}}(m+\rho_{\text{index}}-1)^{m-2}R_0^{m-1}e^{-R_0(m+\rho_{\text{index}}-1)}}{(m-1)!}w^m.$$

In terms of $\widetilde{\mathcal{T}}$ this yields

$$\widetilde{\mathcal{T}}(z;R_0) = \sum_{m=1}^{\infty}\frac{\rho_{\text{index}}(m+\rho_{\text{index}}-1)^{m-2}R_0^{m-1}e^{-R_0(m+\rho_{\text{index}}-1)}}{(m-1)!}\left((1-p)+pz\right)^m. \qquad (8)$$

Unfortunately, we need Taylor expansion around $z = 0$ to derive the state probabilities (and consequently the log-likelihood function). By applying the binomial theorem, $\widetilde{\mathcal{T}}$ can be re-written as

$$\widetilde{\mathcal{T}}(z;R_0) = \sum_{m=1}^{\infty}\frac{\rho_{\text{index}}(m+\rho_{\text{index}}-1)^{m-2}R_0^{m-1}e^{-R_0(m+\rho_{\text{index}}-1)}}{(m-1)!}\sum_{k=0}^{m}\binom{m}{k}(1-p)^{m-k}p^k z^k$$

$$= \sum_{k=0}^{\infty}\sum_{m=k\vee 1}^{\infty}\frac{\rho_{\text{index}}(m+\rho_{\text{index}}-1)^{m-2}R_0^{m-1}e^{-R_0(m+\rho_{\text{index}}-1)}}{(m-1)!}\binom{m}{k}(1-p)^{m-k}p^k z^k$$

$$\widetilde{\mathcal{T}}(z;R_0) = \sum_{k=0}^{\infty}\frac{\rho_{\text{index}}\left(\frac{p}{1-p}\right)^k}{k!}\left(\sum_{m=k\vee 1}^{\infty}\frac{m(m+\rho_{\text{index}}-1)^{m-2}R_0^{m-1}e^{-R_0(m+\rho_{\text{index}}-1)}(1-p)^m}{(m-k)!}\right)z^k,$$

with $m = k \vee 1$ denoting $m = \max\{k, 1\}$.

**Initial estimate for $R_0$**

Since the optimization problem of maximizing the likelihood does not admit a closed-form solution, the ML estimator is obtained with numerical techniques for which a suited initial

estimate for $R_0$ is required. In the following paragraphs we present one possibility for obtaining a useful starting value (which was also implemented and used in our analyses).
Let

$$\overline{\mu} := \frac{1}{I} \sum_{i=1}^{I} \widetilde{t}_i$$

be the observed average chain size (based on a sample of $I$ observed chains $\widetilde{\mathbf{t}}$ like proposed in **Blumberg and Lloyd-Smith, 2013b**). $\overline{\mu}$ represents a reasonable estimate for

$$\overline{\mu} \approx \mathbb{E}\left[\widetilde{T} \mid \widetilde{T} > 0\right] = \frac{\mathbb{E}\left[\widetilde{T}\right]}{1 - \mathbb{P}\left[\widetilde{T} = 0\right]} = \frac{\widetilde{\mathcal{T}}^{(1)}(1; R_0)}{1 - \widetilde{\mathcal{T}}(0; R_0)}.$$

The definition of the skeleton function $\mathcal{K}$ for transmission parameters $\boldsymbol{\omega}$ implies $\mathcal{K}(1; \boldsymbol{\omega}) = 1$. Implicitly deriving **Equation 3** with respect to $z$ implies

$$\mathcal{K}^{(1)}(1; \boldsymbol{\omega}) = \frac{1}{1 - \mathcal{G}^{(1)}(1; \boldsymbol{\omega})},$$

since $\mathcal{G}(1; \boldsymbol{\omega}) = 1$ (just like for any PGF). Moreover, implicitly deriving **Equation 5** with respect to $z$ yields

$$\widetilde{\mathcal{T}}^{(1)}(1; \boldsymbol{\omega}) = p \cdot \mathcal{F}(\mathcal{K}(1; \boldsymbol{\omega}); \boldsymbol{\omega}) + \mathcal{F}^{(1)}(\mathcal{K}(1; \boldsymbol{\omega}); \boldsymbol{\omega}) \cdot \mathcal{K}^{(1)}(1; \boldsymbol{\omega}) \cdot p$$

$$= p \cdot \mathcal{F}(1; \boldsymbol{\omega}) + \mathcal{F}^{(1)}(1; \boldsymbol{\omega}) \cdot \frac{1}{1 - \mathcal{G}^{(1)}(1; \boldsymbol{\omega})} \cdot p$$

$$= p \left( 1 + \frac{\mathcal{F}^{(1)}(1; \boldsymbol{\omega})}{1 - \mathcal{G}^{(1)}(1; \boldsymbol{\omega})} \right).$$

Under the Poisson model, the latter equals to

$$\widetilde{\mathcal{T}}^{(1)}(1; R_0) = p \left( 1 + \frac{\rho_{\text{index}} R_0}{1 - R_0} \right).$$

Next, we can use the first Taylor coefficient of $\widetilde{\mathcal{T}}(z; R_0)$ from **Equation 8**, namely $\widetilde{\mathcal{T}}(0; R_0) \approx e^{-\rho_{\text{index}} R_0}(1 - p)$. In order to obtain a quadratic equation with respect to $R_0$, we further use the approximation $e^{-\rho_{\text{index}} R_0} \approx 1 - \rho_{\text{index}} R_0$, such that $1 - \widetilde{\mathcal{T}}(0; R_0) \approx 1 - (1 - \rho_{\text{index}} R_0)(1 - p)$. This yields the quadratic equation

$$\overline{\mu} \stackrel{!}{=} \frac{p \left( 1 + \frac{\rho_{\text{index}} r_0}{1 - r_0} \right)}{p + (1 - p) \rho_{\text{index}} r_0}$$

with the roots

$$r_0 = \frac{a \pm \sqrt{b}}{c},$$

where

$$a = \rho_{\text{index}}(\overline{\mu}(p - 1) + p) + p(\overline{\mu} - 1)$$
$$b = 4\rho_{\text{index}}(\overline{\mu} - 1)\overline{\mu}(1 - p)p + (\rho_{\text{index}}\overline{\mu} + p - (\rho_{\text{index}} + \overline{\mu} + \rho_{\text{index}}\overline{\mu})p)^2$$
$$c = -2\rho_{\text{index}}\overline{\mu}(1 - p).$$

Should none of the roots lie within $(0, 1)$, we could use the following feature. If the average size of the observed chain equals $\overline{\mu}$, the average size of the complete transmission chains would be roughly $\frac{\overline{\mu}}{p}$ (since the mean value of the binomial distribution $Bin(n, p)$ is $np$). Hence,

$$\frac{\overline{\mu}}{p} \approx \mathbb{E}[T] = \mathcal{T}^{(1)}(1;\boldsymbol{\omega}).$$

*Equation 4* then implies

$$\mathcal{T}^{(1)}(1;\omega) = \mathcal{F}(\mathcal{K}(1;\omega);\omega) + \mathcal{F}^{(1)}(\mathcal{K}(1;\omega);\omega) \cdot \mathcal{K}^{(1)}(1;\omega)$$

$$= 1 + \mathcal{F}^{(1)}(1;\omega) \cdot \frac{1}{1 - \mathcal{G}^{(1)}(1;\omega)}.$$

In case of the Poisson model, the initial estimate for $R_0$ can be therefore obtained by solving the equation

$$\frac{\overline{\mu}}{p} \overset{!}{=} 1 + \frac{\rho_{\text{index}} r_0}{1 - r_0},$$

which has the solution

$$r_0 = \frac{\overline{\mu} - p}{\rho_{\text{index}} p + \overline{\mu} - p}.$$

### Generalized Poisson model

Let $\widetilde{\mathbf{T}} := \{\widetilde{\mathbf{t}}_i\}_{i=1}^{I}$ be a sample of $I$ observed transmission chains where each observed transmission chain $\widetilde{\mathbf{t}}_i$ carries the following information

$$\widetilde{\mathbf{t}}_i := \left(\widetilde{t}_i, \mathbf{x}_i, p_i, \rho_{\text{index},i}\right),$$

namely the observed chain size $\widetilde{t}_i$, the chain characteristics $\mathbf{x}_i$, the probability $p_i$ at which each infection in the chain is observed, and the index case relative transmission potential $\rho_{\text{index},i}$. In the generalized Poisson transmission chain size distribution model we assume that the heterogeneity of the basic reproductive number $R_0$ can be explained by the variability of the demographic characteristics of the transmission chains, namely

$$\log(R_{0,i}) := \boldsymbol{\beta}^T \mathbf{x}_i.$$

The vector $\boldsymbol{\beta}$ describes the effect of the chain characteristics on the basic reproductive number $R_0$ and it is the same for all transmission chains.

To obtain the maximum likelihood estimates for $\boldsymbol{\beta}$, we need initial values of the estimates. One possibility is to use the coefficients from the linear regression model, in which the response values are the individual initial estimates for $R_0$ for each transmission chain. More precisely, imagine that each transmission chain $\widetilde{\mathbf{t}}_i$ is a sample of transmission chains itself and therefore we can obtain the initial $r_{0,i}$ estimates as described above. In the next step, we fit the linear regression model

$$\log(r_{0,i}) := \boldsymbol{\beta}_0^T \mathbf{x}_i + \varepsilon_i, \qquad \varepsilon_i \sim \mathcal{N}(0, \sigma^2),$$

and use $\widehat{\boldsymbol{\beta}}_0$ as the initial values.

## Example: Negative binomial model

Assume that the number of secondary infections caused by an individual is negative binomially distributed with mean $R_0$ and dispersion parameter $\xi$. Its PGF equals

$$\mathcal{G}(z; R_0, \xi) = \left(1 + \frac{R_0}{\xi}(1 - z)\right)^{-\xi}.$$

For the simplicity assume that index case has the same transmission potential as the remaining part of the transmission chain, namely $\mathcal{F} \equiv \mathcal{G}$ (which coincides with $\rho_{\text{index}} = 1$). The skeleton function $\mathcal{K}(z; R_0, \xi)$ is therefore a solution of the equation

$$\mathcal{K}(z;R_0,\xi) = z\left(1 + \frac{R_0}{\xi}(1 - \mathcal{K}(z;R_0,\xi))\right)^{-\xi},$$

which does not admit a closed-form solution. However, as a consequence of the Lagrange inversion theorem its Taylor coefficients around $z = 0$ can be explicitly calculated (**Blumberg and Lloyd-Smith, 2013b**) as

$$\mathcal{K}^{(k)}(0;R_0,\xi) = \frac{\Gamma(\xi k + k - 1)}{\Gamma(\xi k)} \cdot \frac{\left(\frac{R_0}{\xi}\right)^{k-1}}{\left(1 + \frac{R_0}{\xi}\right)^{\xi k + k - 1}}.$$

Since we assumed $\mathcal{F} \equiv \mathcal{G}$, it follows $\mathcal{T}(z;R_0,\xi) = \mathcal{K}(z;R_0,\xi)$ for all $z$. For a transmission chain in which each case is observed with probability $p$, the PGF of the observed transmission chain size equals

$$\widetilde{\mathcal{T}}(z;R_0,\xi) = \mathcal{T}((1-p) + pz;R_0,\xi).$$

By applying the binomial theorem to the Taylor expansion around $z = 0$ of $\mathcal{K}(z;R_0,\xi)$ the higher-order derivatives

$$\widetilde{\mathcal{T}}^{(k)}(0;R_0,\xi) = \sum_{m=k}^{\infty} \frac{\Gamma(\xi m + m - 1)}{\Gamma(\xi m)\Gamma(m - k + 1)} \frac{\left(\frac{R_0}{\xi}\right)^{m-1}}{\left(1 + \frac{R_0}{\xi}\right)^{\xi m + m - 1}} p^k (1-p)^{m-k}$$

are obtained (which coincides with the result from **Blumberg and Lloyd-Smith, 2013a**). In similar manner as in the case of Poisson model, the generalized negative binomial model can be derived by introducing

$$\log(R_{0,i}) = \boldsymbol{\beta}^T \mathbf{x}_i.$$

The sampling density $p$ can vary between the transmission chains (or their characteristics, for instance between the subtypes), while the dispersion parameter $\xi$ is kept constant among all the transmission chains.

