## [Decision Letter]

Thank you for submitting your article "Assessing the danger of self-sustained HIV epidemics in heterosexuals by population based phylogenetic cluster analysis" for consideration by *eLife*. Your article has been reviewed by three peer reviewers, one of whom, Ryosuke Omori (Reviewer #1) served as Guest Editor, and the evaluation has been overseen by Prabhat Jha as the Senior Editor. The following individuals involved in review of your submission have agreed to reveal their identity: Dimitrios Paraskevis (Reviewer #2); Nico Nagelkerke (Reviewer #3).

The reviewers have discussed the reviews with one another and the Reviewing Editor has drafted this decision to help you prepare a revised submission.

Summary:

The authors developed a new method for estimating epidemiological parameters e.g., R0, using viral sequences sampled from patients, via estimating infection tree per each cluster in phylogenetic tree. Using the model they assessed the risk of a generalized heterosexual HIV epidemic in Switzerland. They find that the basic reproduction number is well below 1 they conclude that this risk is negligible.

Essential revisions:

The robustness of the results should be assessed with relaxing the several biological assumptions assumed by the authors. 1) If the first CD4 count (must have been taken before ART initiation) is 310 (median, some even <200 and thus having AIDS) it is questionable whether the assumption of patient becoming non-infectious after 2-2.5 years is realistic. 2) The high acute phase infectivity of HIV has been challenged (e.g. Bellan et al., 2015), this also should be mentioned and considered in the analysis.

Since index cases may differ substantially from other cases (e.g. foreigners in Switzerland without their partner, moving frequently between places), ρ_index_ may well exceed 1.

It is not entirely clear whether – when considering missing patients – the case of a missing index case without "offspring" (i.e., no patient is observed at all) is considered.

The authors assume that the number of secondary infections follows the same distribution regardless of the transmission degree (subsection “Probability generating function of a completely observed uniform transmission chain”). This assumption means that the sexual behaviors are similar among hetero-sexual population whereas the field data of sexual behavior shows huge variation (e.g., Liljeros et al., 2001).

Regarding the estimate for confidence intervals, it is not clear whether the distribution of ML estimator can be approximated by normal distribution or not due to small sample size. If the authors will not apply Wald approximation, the width of confidence interval may change.

The point estimates of R0 shown in the lower panel of Figure 2 (black dot) are likely to be increased since 2009 although the authors' model predicts monotonic decrease. Why does this discrepancy happen? The authors' model predicts HIV will go extinct around 2020, however, this prediction may be optimistic if the predictive model lacks some critical factors.

---

## [Author Response]

Essential revisions:The robustness of the results should be assessed with relaxing the several biological assumptions assumed by the authors. 1) If the first CD4 count (must have been taken before ART initiation) is 310 (median, some even <200 and thus having AIDS) it is questionable whether the assumption of patient becoming non-infectious after 2-2.5 years is realistic. 2) The high acute phase infectivity of HIV has been challenged (e.g. Bellan et al., 2015), this also should be mentioned and considered in the analysis.

We believe that this might be a misunderstanding, since none of the two assumptions was made in our analyses directly but rather used as an intuitive explanation of 1) why the ongoing transmissions are not problematic and 2) to justify the introduction of ρ_index_ (which was already assessed in a sensitivity analysis by setting ρ_index_=1). When verifying the stuttering transmission chains assumption we originally used even an infectious period of 5 years because it is more conservative for this test (longer infectious periods imply more ongoing transmission chains and hence potentially stronger underestimation of R_0_). Despite the minor impact of these assumptions on our method, we further addressed these issues as follows:

Regarding 1) we extended and adjusted the section “Stuttering transmission chains assumption” (now named “Ongoing transmission and stuttering transmission chains assumption”) to assess the role of the duration of infectiousness in our study. Rather than assuming a constant duration, we assumed decreasing periods over time and used the information from the SHCS to estimate the decrease over time. Next, we re-evaluated the conservative maximum number of transmission degrees that had been completed by a certain date (Appendix 1—Figure 3), which were then used in the simulation study to verify the subcritical transmission assumption (now Appendix 1—Figure 5, previously Appendix 1—Figure 3). Additionally, we explicitly assessed the relative bias stemming from ongoing transmission (Appendix 1—Figure 4).

We had addressed point 2) in the original version of the manuscript by performing the sensitivity analysis over a range of ρ_index_ (Appendix 1—Figure 1), where the lower ρ_index_ values implicitly correspond to later immigration to Switzerland and/or the importance of acute phase on transmission, and vice versa – higher ρ_index_ indicates either immigration early during infection or high infectivity long into the chronic phase. We added a more explicit relation between the ρ_index_ values and infectivity in the Discussion. The revised sentence in the Discussion reads:

"Then again, the concrete value chosen may be debatable, especially due to arguable infectivity in chronic phase (studied by Bellan et al., 2015; thus a small ρ_index_ can be caused both by immigration later during chronic infection and by elevated infectivity in the acute phase."

Since index cases may differ substantially from other cases (e.g. foreigners in Switzerland without their partner, moving frequently between places), ρ_index_ may well exceed 1.

This is an excellent point. In principle the index case could exhibit enhanced transmission potential. However, we do not expect that this is often the case and even if the ρ_index_ exceeds 1, this would imply even lower R_0_. We added this comment to the Discussion: "Furthermore, even though theoretically the transmission potential of some index cases could also be enhanced (i.e., ρ_index_>1), for instance for sex workers, we do not expect that this is the case for many transmission chains and would therefore have only marginal effect on our estimates. Besides, since a ρ_index_>1 would lead to even lower R_0_, our main conclusions would not change (in fact, the assumption of ρ_index_<1 is conservative with respect to our conclusion of R_0_<1)." Furthermore, we replaced the "reduced" with "modified" transmission potential/transmissibility throughout the manuscript and extended the range of the sensitivity analyses regarding ρ_index_ beyond 1 (Appendix 1—figure 1).

It is not entirely clear whether – when considering missing patients – the case of a missing index case without "offspring" (i.e., no patient is observed at all) is considered.

We are very grateful for this remark. In our model we indeed considered the transmission chains without any observed cases (i.e., all patients including the index case are missing) by normalizing the state probabilities for the observed transmission chain size distribution by the probability that a transmission chain is observed (namely 1−P[T~=0]). In this sense our model takes into account that the sample of observed transmission chains from the phylogeny is biased, simply because the transmission chains with no observed cases cannot be detected. The same solution was suggested by Blumberg and Lloyd-Smith, 2013b. In the Methods and Materials we added some lines to explicitly explain how the unobserved transmission chains were handled:

"The probability that a transmission chain has observed size of t~≥0 (where t~=0 means that none of the cases of the transmission chain is detected) is given byP[T~=t~]=1t~!T~(t~)(0;R0,ρindex,p).

In particular, the probability that a transmission chain is observed (i.e., the observed size is strictly positive) can be calculated asP[T~>0]=1−P[T~=0]=1−T~(0;R0,ρindex,p).

However, since only the transmission chains with at least one detected case can be extracted from the phylogeny (and therefore to account for the unobserved transmission chains) we are interested in the probability that an observed transmission chain has a specific size. The probability of observing a transmission chain of size t~>0 isP[T~=t~|T~>0]=1t~!T~(t~)(0;R0,ρindex,p)1−T(0;R0,ρindex,p)."

The second source of confusion could be the transmission chains in which the index case (and/or some other intermediaries) are missing. These cases are described with the two assumptions in the Likelihood function section. For instance, if an index case with a single offspring was not sampled while its offspring was detected on the phylogeny, our model would treat this transmission pair as an observed transmission chain of observed size 1 and the sampling density p<1 would account for the 'missing' index case.

The authors assume that the number of secondary infections follows the same distribution regardless of the transmission degree (subsection “Probability generating function of a completely observed uniform transmission chain”). This assumption means that the sexual behaviors are similar among hetero-sexual population whereas the field data of sexual behavior shows huge variation (e.g., Liljeros et al., 2001).

We indeed made the above assumption as a trade-off between accuracy and simplicity, as modelling the human sexual behavior would most likely require changing patterns even within an individual over time. Yet, since our transmission chains are very short (mean 1.19, median 1), we did not expect that the heterogeneity between the individuals within a chain would have an important impact, as we had explained in the Discussion. The following three arguments affirmed this explanation:

• The significance at 5%-level and the direction of different determinants did not change by selecting a random infected individual from each chain to determine the risky sexual behavior of the chain as compared to the main analysis, in which the index case determined the sexual risk behavior of a transmission chain (newly added Appendix 1—figure 6).

• Similarly, the determinants were robust when considering all the follow-up questionnaires (about sex with an occasional partner) from all the patients from a transmission chain to determine the chain’s risky behavior (new Appendix 1—figure 6).

• The comparison between the negative binomial and Poisson based transmission chain size distribution model did not exhibit a strong preference of the former over the latter. Noteworthy, the dispersion parameter of the negative binomial distribution takes into account the heterogeneity between the individuals, hence no significant difference between the models implies that the heterogeneity between the individuals is sufficiently reflected by the variability between the transmission chains in terms of their characteristics (i.e., the model covariables), including the risky sexual behavior (see Comparison between Poisson and negative binomial offspring distribution based models in Appendix 1).

We added a section in the Sensitivity analyses appendix to explain how the effect of the variability in sexual behavior was assessed ("Variation in sexual behavior along transmission chains").

Regarding the estimate for confidence intervals, it is not clear whether the distribution of ML estimator can be approximated by normal distribution or not due to small sample size. If the authors will not apply Wald approximation, the width of confidence interval may change.

We agree with the reviewers that the Wald approximation based on the normal distribution might be debatable. To address this question we did the following:

• We performed parametric bootstrap to assess the assumption of the normal approximation and the performance of the Wald-type confidence intervals in mean of coverage rates. The obtained empirical distribution of the ML estimator could be well approximated by the normal distribution and the coverage rates were very close to or above the target 95% (newly added Appendix 1—figure 10).

• For all the transmission parameters from any of the models presented in our study (newly added Appendix 1 Table 1) we constructed the profile likelihood based confidence intervals, as well as the basic bootstrap confidence intervals (from the parametric and nonparametric bootstrapping) to compare their widths. For our sample size, the Wald confidence intervals turned out to be almost the same as the profile likelihood based confidence intervals, while we did not observe that the Wald-type confidence intervals are systematically wider/narrower than the bootstrap based confidence intervals (Appendix 1—figure 11).

We therefore concluded that the normal approximation Wald-type confidence intervals for our sample size are a reliable and computationally less expensive alternative. We added a section "Confidence intervals" to Appendix 1.

The point estimates of R0 shown in the lower panel of Figure 2 (black dot) are likely to be increased since 2009 although the authors' model predicts monotonic decrease. Why does this discrepancy happen? The authors' model predicts HIV will go extinct around 2020, however, this prediction may be optimistic if the predictive model lacks some critical factors.

This is a very valid remark. In our model, the R_0_ point estimates exhibit a slight increase after 2009, however the number of yet sampled transmission chains from these recent years is still small, which is also reflected by very wide confidence intervals. We admit that the extrapolation up to 2020 might be too optimistic; therefore we modified the plot (Figure 2 and the profiled multivariate plot in Figure 4) such that the R_0_ is now only shown up to year 2015 to avoid misinterpretations. The plot shows the general trend and should be understood as such and not as a definitive prognosis of R_0_ for the future years, as all the models are related to some kind of uncertainty. We also added a note on this issue at the end of the paragraph describing Figure 2 in the Results section: "This extrapolation should be, however, taken with a grain of salt and seen more as a trend rather than a prognosis, since only a few transmission chains have been observed for the recent years (which is reflected by wide confidence intervals)."